# 🦉 OWLS: Scaling Laws for Multilingual Speech Recognition and Translation Models

**William Chen**[1]  **Jinchuan Tian**[1]  **Yifan Peng**[1]  **Brian Yan**[1]  **Chao-Han Huck Yang**[2]  **Shinji Watanabe**[1]

## Abstract

Neural scaling laws offer valuable insights for designing robust sequence processing architectures. While these laws have been extensively characterized in other modalities, their behavior in speech remains comparatively underexplored. In this work, we introduce OWLS, an open-access, reproducible suite of multilingual speech recognition and translation models spanning 0.25B to 18B parameters, with the 18B version being the largest speech model, to the best of our knowledge. OWLS leverages up to 360K hours of public speech data across 150 languages, enabling a systematic investigation into how data, model, and compute scaling each influence performance in multilingual speech tasks. We use OWLS to derive neural scaling laws, showing how final performance can be reliably predicted when scaling. Scaling to larger models can improve ASR performance across the board, in both low and high resource languages, improving the accessibility of speech technologies. Finally, we show how OWLS can be used to power new research directions by discovering emergent abilities in large-scale speech models. Model checkpoints will be released on huggingface for future studies.

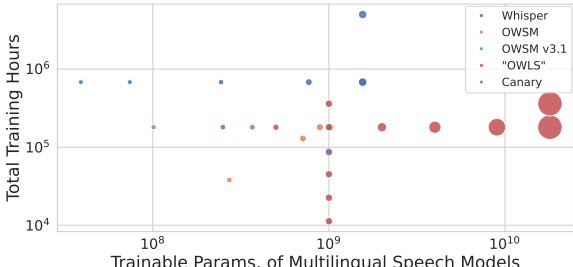

*Figure 1.* **Comparison of previous open models and our OWLS models (red) by parameter count and training dataset size.** Whisper (Radford et al., 2023) and Canary (Puvvada et al., 2024) are trained on *undisclosed* data, while OWSM (Peng et al., 2023b) and the presented OWLS use public data.

## 1. Introduction

Neural acoustic models have shown robust performance in processing human speech information and have demonstrated remarkable capabilities in spoken language tasks (Radford et al., 2023; Peng et al., 2023b; Barrault et al., 2023a). Powered by large-scale training (Baevski et al., 2020; Zhang et al., 2023; Chen et al., 2024; 2022; Li et al.,

2021), Transformer-based (Vaswani et al., 2017) models have dominated the fields of Automatic Speech Recognition (ASR) and Speech Translation (ST).

The state-of-the-art (SOTA) in ASR/ST has now progressed to not only scaling in terms of model and *data size*, but also tasks and *languages*. In recent years, there has been significant interest in developing massively multilingual models that can perform ASR/ST for hundreds, if not thousands, of diverse spoken languages (Chen et al., 2023b; Pratap et al., 2023; Babu et al., 2022; Yu et al., 2023; Chen et al., 2024; Zhang et al., 2023), with the goal of having a single model that can universally convert multilingual speech into text.

However, the architecture of these massively multilingual models is complex, and their scaling properties pose significant challenges for both experimental designs in advancing speech science. This challenge is further exacerbated by the multi-modal nature of spoken language systems, which must handle the complexities of both multilingual text and speech. Prior art on the scaling laws of neural models deviates significantly from the goal of SOTA universal systems. The majority study single-task and single-modality systems (Biderman et al., 2023; Ghorbani et al., 2022; Zheng et al., 2022), while multilingual work concentrates only on settings where a few languages are supported (Fernandes et al., 2023; Yang et al., 2023; Li et al., 2021).

To address this, we present OWLS, a **O**pen **W**hisper-style

[1]Carnegie Mellon University [2]NVIDIA. Correspondence to: William Chen <williamchen@cmu.edu>, Chao-Han Huck Yang <hucky@nvidia.com>, Shinji Watanabe <shinjiw@ieee.org>.

**L**arge-scale neural model **S**uite for Speech Recognition and Translation. OWLS contains 13 fully transparent[1] speech foundation models for ASR/ST, pre-trained on up to 360K hours of multilingual data across 150 languages, with each model ranging from 0.25B to 18B parameters (Figure 1). We experiment with scaling in terms of both model and data size, and analyze the change in downstream ASR/ST performance. Through these investigations, we derive a neural scaling law to predict the change in model performance for each task and language. We also evaluate *test-time* capabilities of large-scale ASR/ST models, studying how new abilities emerge at scale and showing how speech model scaling can be benefits to new languages with in-context learning. Our contributions are summarized as follows:

- We open-source OWLS, a collection of 13 Whisper-style ASR/ST models trained on up to 360K hours of publicly available data and 150 languages. We will also release all model training code, training logs, and intermediate checkpoints.

- We train and release an OWLS model with 18B total parameters, which makes it the largest of all publicly known ASR/ST models and nearly double that of prior work (Zheng et al., 2022).

- We systemically evaluate the effects of model and data scaling on ASR and ST, developing the first set of neural scaling laws for these tasks. We not only measure the usefulness of model scaling, but also identify failure cases that it is not able to overcome.

- We evaluate the test-time capabilities of frozen large-scale speech foundation models via in-context learning, and discover several new emergent abilities present in large models that are absent in smaller ones.

## 2. Background and Related Work

### 2.1. Neural Scaling Laws
Previous research has shown that the performance of Transformer-based (Vaswani et al., 2017) models at scale can be empirically predicted with three fundamental variables: the model size $N$, the training data size $T$, and the compute budget $B$ (Hestness et al., 2017; Rosenfeld et al., 2020; Kaplan et al., 2020; Hernandez et al., 2021; Ghorbani et al., 2022; Fernandes et al., 2023). This can be summarized by modeling the change in the cross-entropy loss $L$ when varying each variable independently:

$$L(x) = L_\infty + \beta_x x^{\alpha_x}, \qquad (1)$$

where $x \in (N, T, B)$, $L(x)$ is the reducible loss that obeys the power-scaling law, and $L_\infty$ is irreducible loss. $\beta$ and

$\alpha$ are thus the empirically learned variables of the power law. Varying the value of $x$ allows a practitioner to estimate the scaling behavior in different settings. When $x = N^2$, for example, the power law models the data-rich ($T \to \infty$) and compute-rich ($B \to \infty$) setting. Previous work (Gu et al., 2023) in language model re-scoring has shown that the Word Error Rate (WER) can also be modeled as a power law function of $x$. We can thus modify Equation 1 as follows:

$$\text{WER}(x) = \beta_x x^{\alpha_x}. \qquad (2)$$

We empirically show that this power law can also generalize to the multi-modal task of ASR (Figures 3 and 9), allowing true downstream performance to be easily predicted when $x = N, B$. Furthermore, we also observe that it can be applied to ST (via $\text{BLEU}(x) = \beta_x x^{\alpha_x}$) and thus extends our findings to more tasks (Figures 7 and 6).

### 2.2. Scaling Laws for text and vision
The impact of scaling neural models has been thoroughly studied in the domains of text and vision. Early studies in scaling text models focused on supervised tasks such as machine translation (MT) (Gordon et al., 2021; Ghorbani et al., 2022). The most relevant work to ours is from Fernandes et al. (2023), who devised scaling laws for multilingual MT models. However, these are only trained on two translation tasks/languages. In comparison, our work evaluates on over 100 languages and tasks.

Later studies focused instead on scaling self-supervised LLMs (Biderman et al., 2023; Tay et al., 2023; Kaplan et al., 2020). Kaplan et al. (2020) empirically showed that language modeling obeys a power law w.r.t $x = N, T$, and $B$. Biderman et al. (2023) released a suite of open-access LLMs, and showed how they can be used to understand scaling behaviors on downstream tasks. Our research can be viewed as a combination of these works, albeit applied to speech: we introduce a suite of open-access large ASR/ST models and also derive scaling laws for downstream tasks.

In vision, there is existing literature on the scalability of vision encoders on image classification tasks (Zhai et al., 2022). However, these tasks do not require multi-modal understanding. Our work is thus most similar to those on text-to-image/image-to-text tasks (Henighan et al., 2020). However, we focus on the speech modality while also considering multi-tasking and zero-shot behaviors.

### 2.3. Multilingual Processing and Scaling in Speech
Multilingual ASR is the concept of having a single model that can recognize speech in many languages (Watanabe

---

[1]We follow the definition of "transparency" (Dabbish et al., 2012) on open-source, open-data, and open transcripts.

[2]We assume that the model parameters are equally distributed between the encoder and decoder for encoder-decoder architectures. Otherwise, the law can also be formulated as a bivariate function w.r.t. to the encoder parameters $N_e$ and decoder parameters $N_d$ (Fernandes et al., 2023; Ghorbani et al., 2022)

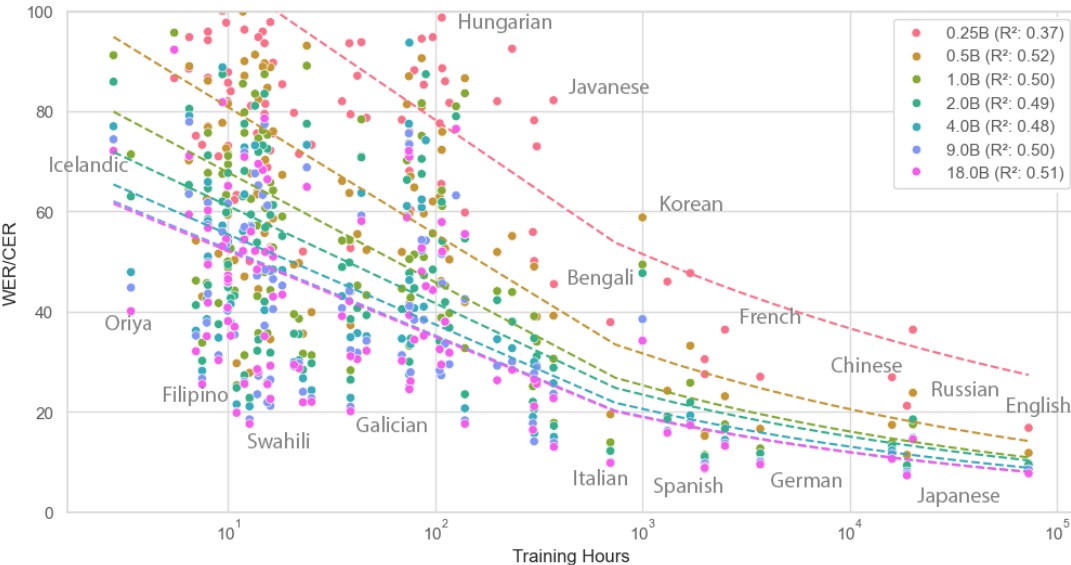

*Figure 2.* **The effect of scaling model size on the 102 FLEURS languages, plotted as WER (or CER) versus available training data.**
Although WER/CER generally decreases with more training data, the relationship is only moderately correlated, as indicated by the $R^2$
values in the legend. Model performance is also influenced by domain alignment and orthographic transparency: for instance, more
transparent languages (e.g., Spanish, Italian) often achieve lower error rates with less data than opaque languages (e.g., English, French).

et al., 2017a). While initial investigations focused on only
combining a few languages together (Conneau et al., 2021),
modern multilingual ASR models are capable of handling
hundreds, if not thousands, of languages (Zhang et al., 2023;
Pratap et al., 2023; Chen et al., 2024; Radford et al., 2023; Li
et al., 2022). Recent SOTA multilingual speech models have
begun supporting tasks in addition to ASR. Joint language
prediction and speech recognition is now a common method
of developing multilingual ASR models (Chen et al., 2023b;
Radford et al., 2023). Whisper-style models (Radford et al.,
2023; Peng et al., 2023b) use a system of language and
task prompts to also perform language identification, speech
translation, and timestamp prediction. On the other hand,
the Seamless family (Barrault et al., 2023a;b) leverages task
decomposition to perform ASR within a speech-to-speech
translation framework. Our work focuses on Whisper-style
models, as their use of task prompts allow us to easily eval-
uate the effects of scale on zero/few-shot performance.

There have been few studies on neural scaling laws for
speech. Droppo & Elibol (2021) and Cuervo & Marxer
(2024) devised neural scaling laws for self-supervised acous-
tics models and speech language models, respectively. How-
ever, their evaluations are limited to simple probes due to
the text-less nature of these models, and cannot be easily
applied to typical speech tasks. Zheng et al. (2022) and Li
et al. (2021) experimented with scaling monolingual and
multilingual models respectively to 10B parameters, but the
models are trained only on internal data and remain unre-
leased. Neither works attempt to devise empirical scaling
laws nor study the enhanced capabilities of larger models.

## 3. The OWL Suite

### 3.1. Dataset

We largely rely on the OWSM v3.2 (Tian et al., 2024) dataset
for our experiments. It consists of 180K hours of ASR/ST
data gathered across 25 public corpora, covering 150 unique
languages. For our experiments on scaling up the training
data size beyond 180K hours, we also include an additional
180K hours from a cleaned subset of YODAS (Li et al.,
2023) from Peng et al. (2025), for a total of 360K hours.
Note that this YODAS data is only used to train two models
(OWLS 1B 360K and OWLS 18B v2). More details about
the dataset can be found in Section A in the Appendix.

### 3.2. Training Details

All OWLS models follow a Transformer (Vaswani et al.,
2017) encoder-decoder architecture trained using a hybrid
CTC/attention (Graves et al., 2006; Watanabe et al., 2017b)
loss. The inputs to the Transformer are 80-dimension log-
Mel filterbanks extracted with a frame shift of 10ms, which
are then down-sampled 4 times by a stack of convolution
layers. The prediction targets are text tokens with a 50K sub-
word vocabulary (Kudo, 2018). We also use Whisper-style
training (Radford et al., 2023): all utterances are padded
to 30 seconds, and the model is jointly trained to perform
language identification, ASR, ST, and timestamp prediction.

We conduct our experiments with the ESPNet (Watanabe
et al., 2018) toolkit. Since our goal is a systematic study
of large-scale speech models, we take an experimental ap-
proach similar to Biderman et al. (2023): we design our

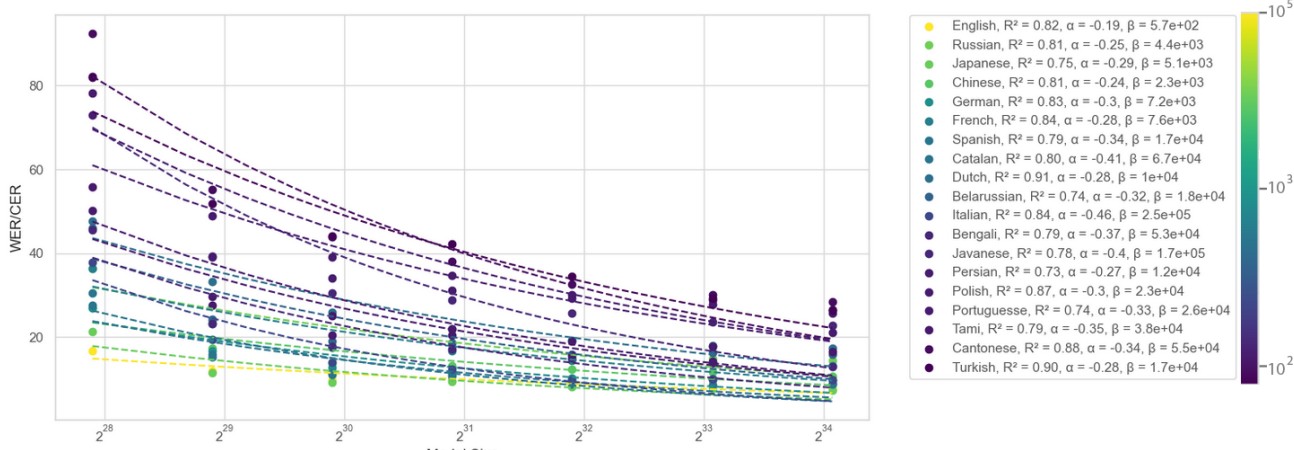

*Figure 3.* **The effect of model scaling on WER/CER on FLEURS.** Languages are color-coded by the amount of training data. For readability, we only show the top-20 languages (by data amount) in our training corpus. We find that model scaling is consistently predictive of downstream WER/CER across languages. Scaling curves for other languages can be found in Figure 14 in the Appendix.

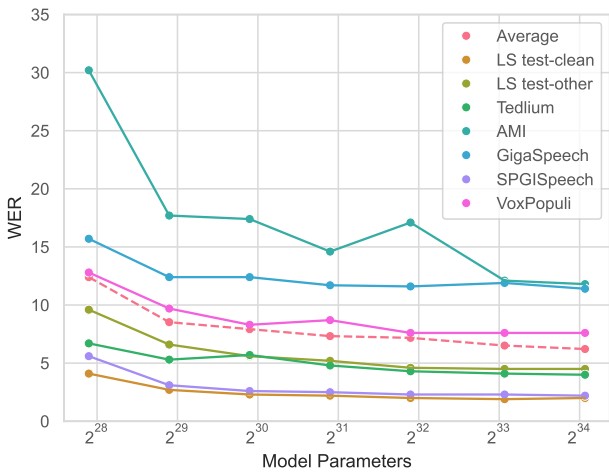

*Figure 4.* **WERs on multi-domain English ASR by model size.**

experiments to prioritize training stability and controllability over squeezing out the best possible performance. We therefore use *the exact same hyper-parameters* for all models, varying only the data or model size to fit the appropriate scaling experiment. More details on training can be found in Appendix B.

## 4. Pre-Training Experiments

### 4.1. Scaling Model Size

We experiment with scaling the model parameters of the OWLS models from 0.25B to 18B parameters, roughly doubling the total model parameters with each iteration. This leads to a total of 7 model sizes (0.25B, 0.50B, 1B, 2B, 4B, 9B, 18B). For each model size we scale the depth and width of the encoder and decoder in tandem, while allocating the

model parameters equally between both. More details about each model can be found in Appendix B.

**Multilingual ASR:** To evaluate the multilingual performance of the OWLS models, we use the 102-language FLEURS test set (Conneau et al., 2022). Figures 2 and 3 show WER for different languages as a function of per-language training data size and model size respectively, and measure their correlation with WER using the co-efficient of determination, $R^2$. We find that model scaling consistently improves WER/CER of each language across all data levels (Figure 2). However, the *amount of data used for any given language is only somewhat predictive of its WER/CER* ($R^2 \simeq 0.5$, Figure 2). In other words, *we cannot easily fit a language-agnostic data scaling law*. This of course, is expected. Some languages are naturally more difficult to model for ASR than others (i.e Spanish vs Chinese) (Taguchi & Chiang, 2024), so they will require more training data. On the other hand, *language-specific model size scaling laws are highly predictive of WER/CER* ($R^2 \simeq 0.95$, Figure 3). Finally, we want to highlight the significant improvement on WER in low-resource languages when scaling to larger model sizes. The average WER on the 50 lowest-resource languages (less than 35 hours of training data) in our dataset decreases from 59 to 45 when model size increases from an already large size of 1B to 9B. *Larger models can improve ASR performance across the board, in both low and high resource languages.*

**Multi-domain ASR (English):** We test robustness of OWLS models to different data domains by evaluating on 6 standard ASR benchmarks: AMI (Carletta, 2007), LibriSpeech (Panayotov et al., 2015), SPGISpeech (O'Neill et al., 2021), Tedlium (Hernandez et al., 2018), VoxPopuli (Wang et al., 2021b), and GigaSpeech (Chen et al., 2021).

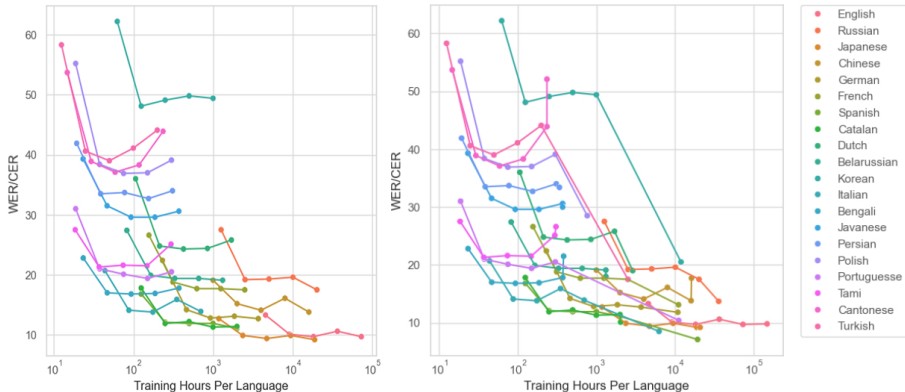

*Figure 5.* **The evolution of FLEURS WER/CER for the top 20 languages by data size, as more training data is added for each language and given a fixed model capacity.** *Left*: impact on WER/CER when scaling from 11K to 180K total hours, when all data is from the same distribution. *Right*: impact on WER/CER from adding in data from a new domain/distribution (YODAS), when further scaling from 180K to 360K total hours. Plots for more languages can be found in Figure 15 in the Appendix.

Figure 4 shows the results of these experiments. *ASR improves significantly with scale across all domains*, with the average WER almost halving from 12.1 to 6.3 when scaling from 0.25B to 9B parameters. *The effects of scale are apparent even when going beyond the typical maximum ASR model size of 2B parameters*, with a relative reduction in WER of 11.3% when scaling from 2B to 9B.

**Translation:** We study the effects of parameter scaling on English to X and X to English translation. The results are shown in Figures 7 and 6 respectively. We observe that scaling the model parameters leads to significant improvements in BLEU scores for all languages. This observation holds true even for high-resource language pairs. For high-resource English to German, *scaling from an already large 1B model to a 9B variant nearly doubles the BLEU score from 16.6 to 28.9* (Figure 7). Figure 7 also shows that *some models are too small to functionally perform ST:* the 0.25B OWLS model is unable to produce intelligible output (BLEU < 5) for 9 of the 15 English to X translation pairs. In comparison, the 9B OWLS model functions reasonably well (BLEU > 15) on 12 of the 15 pairs.

However, there are also limitations of model scaling. Figure 6 shows the effects of scaling on X to English ST. While 4 out of the 5 language pairs show improvement trends similar to Figure 7, the BLEU scores for Japanese do not increase significantly. Importantly, there is only 1 hour total of Japanese to English ST to English data in the OWLS training corpus. We can thus conclude the following: *while parameter scaling can significantly improve ST performance, it cannot overcome cases where there is inherently insufficient amounts of data to learn the task.*

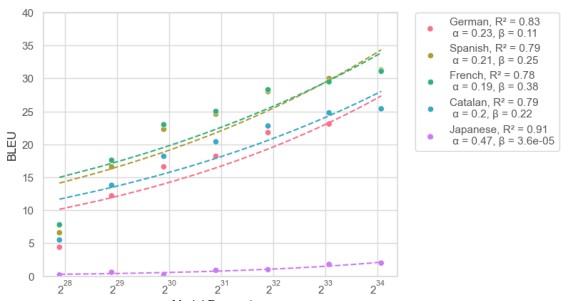

*Figure 6.* **BLEU scores on X to English speech translation.**

### 4.2. Scaling Data Size

We evaluate how varying the amount of data used to train an OWLS model can affect downstream performance. To do so, we first create smaller training splits by uniformly downsampling the 180K hour base training set by 50%, 25%, 12.5%, and 6.25%. We also experiment with using a larger amount of data by collecting an additional 180K hours from YODAS (Section 3.1). For these experiments, we fix the model size at 1B parameters. This leads to a total of 6 different models trained on 360K, 180K, 90K, 45K, 22.5K, and 11.25K hours of speech respectively. We use an evaluation protocol similar to the one in Section 4.1, benchmarking the model on Multilingual ASR and ST.

**Multilingual ASR:** Figure 5 (left) shows the effect of data scaling on the WER of each language from 11.25K to 180K hours, given a fixed model capacity. While a training set generally leads to better performance for most languages, we also observe degradations in WER/CER for some, likely due to interference from similar languages (e.g. Chinese interference for Cantonese). Figure 5 (right) shows the impact of adding in data from a new domain/distribution (YODAS) when scaling from 180K hours to 360K hours. With the addition of 180K hours of high quality data from YODAS,

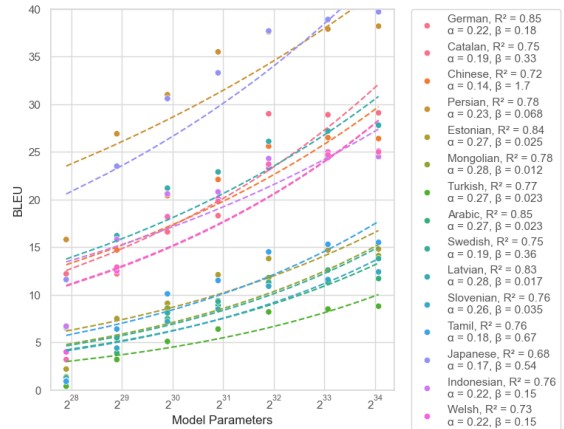

Figure 7. **BLEU scores on English to X speech translation.**

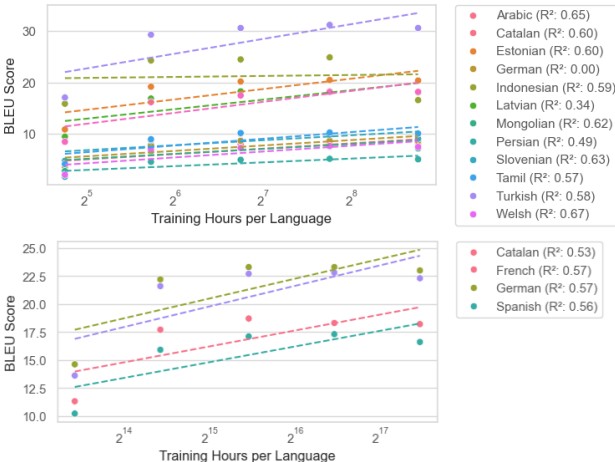

Figure 8. **BLEU scores on EN to X (*top*) and X to EN (*bottom*) ST with different dataset sizes.**

many languages with saturated performance when scaling from 22K to 180K hours (Korean, Polish, Dutch) experience large improvements in WER/CER. Our findings can thus be summarized as the following: *data scaling without additional diversity leads to quickly saturated performance.*

**Translation:** Similar to our findings in Figure 2, we find that ST data quantity is only loosely correlated with downstream performance ($R^2 \simeq 0.55$). The top and bottom portions of Figure 8 show the change in BLEU score as the training data size increases for English to X and X to English, respectively. While BLEU score is positively correlated with a larger dataset size for most translation pairs, we also observe significant degradations in English to German (Figure 8). We hypothesize that this may be due to the 1B model's limited capacity as data size increases, but leave more concrete analyses to future work. Finally, we note that we exclude results from the 360K model in this analysis, since the additional 180K hours from YODAS did not contain any ST data.

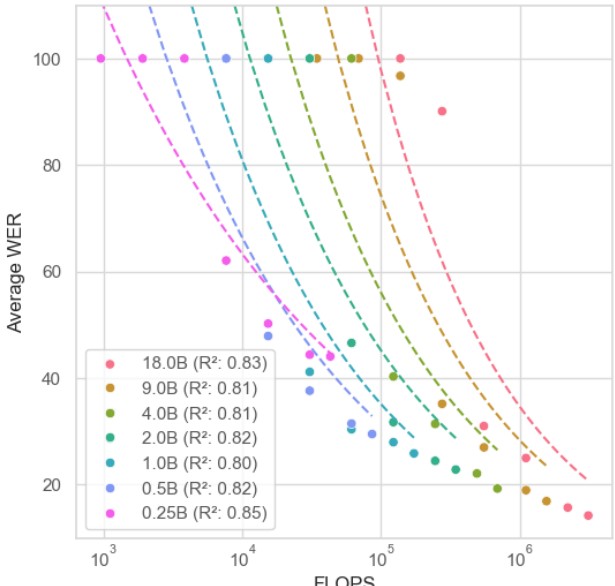

Figure 9. **Average multilingual WER for each model size throughout different stages of training.**

### 4.3. Scaling Compute

Another method of evaluating the effects of scaling is by predicting the test WER as a function of the FLOPS used for training. This allows models to be evaluated in the compute-equivalent setting and considers the fact that larger models will take longer to train. To model this relationship, we test OWLS models of various sizes on FLEURS with various intermediate checkpoints. Specifically, we perform inference with each model after 15K, 30K, 60K, 120K, 240K, 480K, and 675K training steps. The training TFLOPS of each checkpoint is then calculated by profiling each model size for a few steps and scaling the result to 15K-675K steps. We only evaluate on English and two other randomly chosen languages (Spanish and Turkish) to reduce computing costs. Figure 9 shows the evolution of average WER from the 3 languages for each model size as training progresses. We find that for a fixed parameter size, the WER of the final checkpoint can be reliably predicted as a function of the training compute ($R^2 \simeq 0.82$). This means that *one can reasonably predict the final WER of the model given the WERs of initial checkpoints*. As expected, smaller models are more compute efficient, being able to reach a much lower WER with lower TFLOPS spent.

### 4.4. Further Scaling

We combine our findings in model and data scaling to make a preliminary exploration in further scaling OWLS models. We scale an 18B parameter OWLS model to 360K hours of data, which we designate as OWLS 18B v2. We compare this model with other OWLS models and SOTA ASR models

*Table 1.* **WER/CER of OWLS models vs SOTA ASR models on various benchmarks**: FLEURS, AISHELL (zh-CN), LibriSpeech test-clean (eng), ReazonSpeech (jpn), and Ksponspeech (kor). Canary is only trained on 4 European languages. OWLS models perform comparably, if not better than, models like Whisper Large v3 that are trained on much more data (14x more in the case of Whisper).

| Model | FLEURS (eng) | LS clean | FLEURS (jpn) | Reazon | FLEURS (zh-CN) | AISHELL | Kspon |
|---|---|---|---|---|---|---|---|
| Canary | 7.1 | **1.5** | - | - | - | - | - |
| Whisper v3 | **4.1** | 2.0 | **4.9** | 15.1 | 7.7 | 5.1 | **13.4** |
| Qwen2Audio | 9.4 | 1.6 | 20.1 | 50.0 | **7.5** | 8.7 | 53.3 |
| SenseVoice S | 10.3 | 3.2 | 13.1 | 37.1 | 9.6 | 3.0 | 24.5 |
| SenseVoice L | - | 2.6 | - | - | - | 2.1 | - |
| Seamless M | 8.3 | 4.2 | 15.9 | 34.9 | 15.7 | 9.6 | 27.3 |
| Seamless L | 7.3 | 3.7 | 17.6 | 36.6 | 17.0 | 8.7 | 32.4 |
| OWLS 1B | 9.7 | 2.3 | 9.2 | 7.8 | 13.8 | 6.2 | 17.5 |
| OWLS 9B | 8.5 | 1.9 | 7.7 | 7.3 | 11.6 | **4.8** | 15.8 |
| OWLS 18B | 7.7 | 2.0 | 7.2 | 7.5 | 10.6 | **4.8** | 15.2 |
| OWLS 18B v2 | 6.8 | 2.0 | 6.7 | **7.2** | 10.1 | **4.8** | 15.0 |

*Table 2.* **WER on Librispeech test-other when using greedy search (left) and balancing test-time compute budget (right).** We exclude 0.5B and 1B OWLS models since there is no beam size that consumes ∼40-50 TFLOPS.

| Params. | Beam Size | TFLOPS | WER |
|---|---|---|---|
| 0.25B | 1 / 10 | 1.3 / 48.7 | 9.6 / 8.3 |
| 2B | 1 / 4 | 11.1 / 36.2 | 5.2 / 4.7 |
| 4B | 1 / 2 | 23.4 / 42.3 | 4.6 / 4.5 |
| 9B | 1 | 47.7 | 4.5 |

(Radford et al., 2023; Puvvada et al., 2024; Barrault et al., 2023b; Chu et al., 2024; An et al., 2024) in Table 1. OWLS 18B v2 obtains the best or second best result on 5 of the 7 test sets, performing comparably if not better than models trained on more data, like Whisper and Qwen2Audio.

## 5. Test-Time Experiments

### 5.1. Beam Search

One advantage of smaller models is the ability to leverage more complex decoding algorithms during inference. For larger models, using these techniques would be unfeasible within GPU memory constraints. To make the performance more fair at the compute-level, we conduct analyses where all models have the same fixed test-time compute budget. Smaller models may leverage beam search with larger beam sizes, while larger ones may be constrained to only greedy decoding. Table 2 shows the WER on LibriSpeech test-other when test-time compute is balanced at ∼40-50 TFLOPS across the 0.25B, 2B, 4B, and 9B OWLS models. We note that the 0.5B, 1B, and 18B OWLS models are excluded since there is no beam size that consumes a similar number of TFLOPS. We first find that the WER of all models are sensitive to beam size, albeit with diminishing returns as model size increases. Even when using equivalent compute,

larger models clearly perform better than smaller models at test-time (4.5 WER for 9B vs 8.3 WER for 0.25B). This shows the viability of large-scale ASR models in production settings. Additional multilingual results can be found in Appendix C.

### 5.2. Emergent Ability

LLMs are shown to exhibit drastically improved performance on certain tasks as the model size increases, even if the training data remains unchanged (Wei et al., 2022). In this section, we study if large-scale ASR models can also exhibit these "emergent abilities[3]". We focus on three abilities that we newly discover: orthographic understanding, code-switching, and mondegreens. Results for contextual biasing, the first known example of emergent abilities in ASR models (to our knowledge), are found in Appendix G.

**Orthographic Understanding:** Orthographic transparency describes the relationship between the phonetics (sounds) of a language and its written form. Opaque languages (e.g. Chinese and Japanese) have complex many-to-one or one-to-many relationships from sound to symbol, making ASR particularly difficult (Taguchi & Chiang, 2024). Examples of this phenomena are shown in Table 3. We hypothesized that larger OWLS models will exhibit enhanced robustness to orthographic opacity. To measure this, we calculate the normalized CER (N-CER) by normalizing all symbols to a single orthography. This can then be compared to the unnormalized CER. A model with a good N-CER but poor CER has strong phonetic capabilities but poor orthographic

---

[3]In our work, we define "emergent abilities" as those exhibited by larger models and not by smaller models. Wei et al. (2022) originally used a stricter definition where emergent abilities as those that can not be extrapolated from scaling curves. However, Schaeffer et al. (2023) later showed that the emergence can in fact be predicted with finer-grained evaluation metrics.

*Table 3.* **Orthographic opacity examples of Japanese and Chinese.** The same phone sequence can be written in different ways.

| Orthography | Example |
| --- | --- |
| Romanization (zh) | shì shī shì |
| Simp. Chinese | 室诗士 |
| Trad. Chinese | 室詩士 |
| Romanization (jp) | hashi |
| Hiragana | はし |
| Katakana | ハシ |
| Kanji | 橋 |

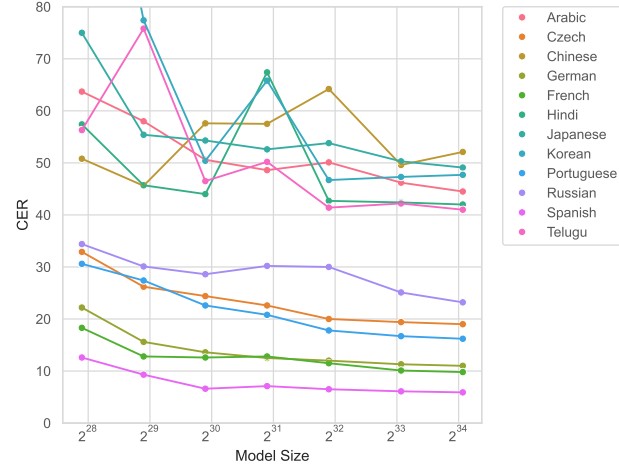

*Figure 11.* **CER on zero-shot English-X code-switching.**

2023a). We collect an evaluation set of bilingually code-switched English for 12 languages from Yan et al. (2025) and test OWLS models of different sizes. Figure 11 shows the results on each code-switched language. *We find that scaling can lead to significant reductions in code-switched CER, but the benefits are unevenly distributed.* Many of the improvements lie in languages that also use the Latin alphabet, like Portuguese, while languages with very different orthographies (such as Chinese) see no improvement. More details about the data are in Appendix D.

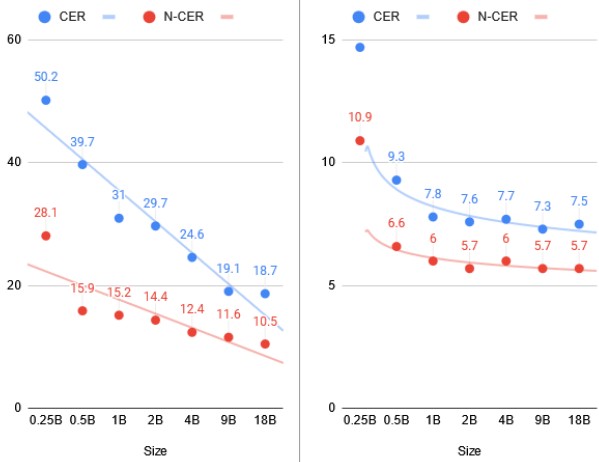

*Figure 10.* **Effects of model scaling on orthographic understanding on Chinese (left) and Japanese (right)**. The quick saturation in N-CER shows that scaling does not have a large effect on the *phonetic* understanding in ASR models. However, the raw CER trend shows that large-scale models exhibit significantly stronger *orthographic* capabilities.

understanding. Models are tested on Taiwanese Chinese Mandarin (zh-TW) and Japanese (Figure 10). The N-CER curve shows that scaling does not have a large impact on learning phonetics: *small models already exhibit strong performance in phonetically mapping speech to text*. On the other hand, the steeper CER curve calculated from the raw model outputs indicate that *larger models exhibit significantly stronger orthographic capabilities*. Another key finding in this experiment was the overall robustness of larger models to zh-TW, which is a minority dialect relative to Mainland Chinese (zh-CN). *Larger models are much more capable of providing fair performance across both dialects* (see Table 1 for zh-CN scores), which aligns with the findings in Section 4.1 on low-resource languages.

**Code-switching:** In multilingual societies, it is common for more than one language to be spoken within a single utterance. However, despite multilingual training, most existing ASR models are incapable of accurately recognizing code-switched speech in a zero-shot manner (Peng et al.,

**Mondegreen:** Humans are capable of constructing semantically meaningful sentences from mis-recognized speech (such as mishearing "José, can you see" from "O say can you see"). This phenomena is known as a mondegreen. We hypothesize that large ASR models learn more semantic mappings than smaller ones, enhancing their ability of constructing mondegreens. We evaluate this technique by purposefully providing the model an English ASR task token along with speech from 3 non-English languages from FLEURS. The generated text is then evaluated by using the perplexity of a pre-trained OPT 2.7B LLM (Zhang et al., 2022b), such that a lower perplexity corresponds to a semantically plausible English sentence for humans. To ground these numbers, we also perform a qualitative analysis with 13 human volunteers, who provided a mean opinion score (MOS) on the semantic coherence for each generation on a scale from 1 to 5 (higher is better). The results of the mondegreen evaluations are shown in Table 4. We observe that larger models obtain consistently better perplexity scores across all model sizes. Similarly, we also find that higher MOS scores trend well with model size. This suggests that *larger ASR models are indeed more capable of "mis-hearing" in a semantically sound manner*. While this phenomena is likely an artifact of scaling that does not directly relate to WER, we believe that such findings may lead to more research on the

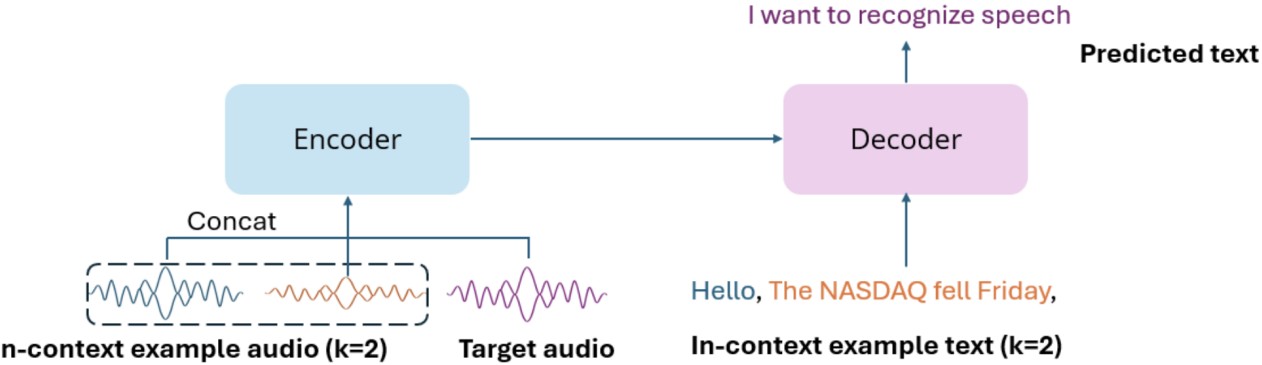

*Figure 12.* **The ICL inference process.** The speech of the in-context examples and test audio are concatenated along the sequence dimension. The text of the in-context examples are also concatenated and used as a prompt for the decoder.

*Table 4.* **Evaluation of mondegreen capabilities.**

| Params. | PPL | MOS |
|---------|------|-----|
| 0.25B | 1338 | 1.9 |
| 0.50B | 728 | 4.1 |
| 1B | 559 | 3.5 |
| 2B | 491 | 3.6 |
| 4B | 436 | 3.8 |
| 9B | **372** | **4.8** |
| 18B | 429 | 4.4 |

*Table 5.* **Quechua CER on ICL with 0 / 1 / 2 / 3 examples.** The overall best result is **bolded** while the best result for each model size is underlined.

| Params. | $k=0$ | $k=1$ | $k=2$ | $k=3$ |
|---------|-------|-------|-------|-------|
| 0.25B | 36.9 | 35.1 | 33.7 | 34.5 |
| 0.50B | 53.3 | 39.2 | 33.8 | 33.9 |
| 1B | 41.8 | 35.0 | 31.6 | 31.8 |
| 2B | 47.3 | 35.1 | 31.9 | 33.2 |
| 4B | 40.4 | 32.4 | 31.2 | 31.8 |
| 9B | 38.3 | 31.3 | 28.1 | **27.4** |
| 18B | 41.3 | 32.7 | 31.3 | 28.1 |

properties of neural networks that emerge at scale, and how they relate to the human perceptions of spoken language. More experimental details and sample inputs/outputs are in Appendix F and `https://wanchichen.github.io/owls-samples`, respectively.

### 5.3. In-Context Learning of OWLS

LLMs are capable of few-shot task performance via in-context learning (ICL) (Brown et al., 2020). Large-scale ASR models like Whisper have shown potential in performing ICL, albeit with very limited capabilities. In this section, we evaluate if the ICL ability of OWLS models improve as the model size scales. To do so, we evaluate the model on ASR for a language unseen during training. We provide the model with 0 to 4 in-context examples to benchmark its ability to learn at test-time. We use Quechua as the unseen language, with data sourced from the Siminchik (Cardenas et al., 2018) corpus. We perform ICL using the same $k$-NN approach as Wang et al. (2024a), where $k$ utterances with the lowest Euclidean distance (when embedded by the encoder) from the target speech are selected from the training set as in-context examples. The audio from the in-context examples are concatenated with the target speech, while the concatenated text examples are fed as an input prompt (Figure 12). We find that while all model sizes are capable of using in-context examples in some capacity, only **the largest models** (9B and 18B) can take advantage of all

*three* in-context examples (Table 5). For the 4B and smaller models, performance degrades when using more than *two* in-context examples. Sample outputs and further details can be found in Appendix H.

## 6. Conclusion and Future Work

This paper introduces OWLS, a suite of 13 joint ASR/ST models designed to help researchers understand the scaling behaviors of multi-modal, multi-language, multi-task models. OWLS models range from 250M to 18B parameters, trained on 11K to 360K hours of speech. In fact, the 18B OWLS model is the largest speech model in known literature. With OWLS, we show that the affects of scaling parameter, training data, and compute can lead to reasonable direct predictions of downstream ASR/ST performance. We also study the emergent capabilities of large-scale ASR/ST models, showing for the first time how larger speech models exhibit stronger in-context abilities and understanding of human language. In the future, we plan to (i) scale model training to even larger datasets and more diverse tasks, and (ii) investigate more scaling effects for adaptation, while also developing new benchmarks to better understand the emergent capabilities of spoken language models with open and diverse research communities together.

## Acknowledgments

Parts of this work used the Bridges2 at PSC and Delta/DeltaAI NCSA computing systems through allocation CIS210014 from the Advanced Cyberinfrastructure Coordination Ecosystem: Services & Support (ACCESS) program, supported by National Science Foundation grants 2138259, 2138286, 2138307, 2137603, and 2138296.

## Impact Statement

This paper presents OWLS, a suite of open-access, reproducible, large-scale joint ASR and ST models. Unlike most other ASR foundation models at this scale, all of the models in this work are trained on publicly accessible datasets and open-source codebases. To facilitate reproducibility, we will also release all intermediate checkpoints, optimizer states, and the final model checkpoint. Our goal is to provide researchers with additional resources and artifacts to better understand the scaling properties of large-scale speech models. We also offer detailed breakdowns of computational resources and costs in the Appendix.

### Societal Consequences

There are many potential societal consequences of machine learning, most of which we will not highlight here because they are common across the entire field. Instead, we will discuss the aspect of our work that is most unique: the impact on society resulting from model scaling. Training the OWLS models required many GPUs, which can consume large amounts of electricity. Although our computing costs are insignificant compared to those incurred in LLM training (*i.e.*, we use at most 48 GPUs at once), they remain large relative to most other work.

### Ethical Aspects

Our models, like all machine learning models, are prone to bias due to uneven distributions in the training data. Although we show that model scaling can lead to more fair performance across different languages, it can still be prone to hallucinations and generate incorrect output.

A portion of the training data that we use was accessed under non-commericial licenses. To follow the spirit of these datasets' access conditions, all of our models are also released under non-commercial licenses. We emphasize that the models are released for research purposes and discourage use outside of this original use-case.

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

*Table 6.* Overview of datasets used in the 180K OWSM v3.2 dataset. The language column indicates the language used in monolingual datasets and the number of languages in multilingual datasets.

| Dataset | License | Language(s) | Domain | Hours |
|---|---|---|---|---|
| MLS (Pratap et al.) | CC BY 4.0 | 8 | Audiobook | 44K |
| WeNetSpeech (Zhang et al., 2022a) | CC BY 4.0/SA | Mandarin | Variety | 22K |
| Russian Open STT (Slizhikova et al., 2020) | CC-BY-NC | Russian | Variety | 20K |
| Reazonspeech (Yin et al., 2023) | Apache 2.0 | Japanese | Television | 15K |
| Common Voice 13 (Ardila et al., 2020) | CC0-1.0 | 92 | Read | 13K |
| GigaSpeech (Chen et al., 2021) | Apache 2.0 | English | Variety | 10K |
| GigaST (Ye et al., 2023) | CC BY NC 4.0 | 2 | Variety | 24K |
| MuST-C (Cattoni et al., 2021) | CC BY NC ND 4.0 | 16 | Talk | 10K |
| CoVoST2 (Wang et al., 2021a) | CC BY NC 4.0 | 22 | Read | 8550 |
| SPGI (O'Neill et al., 2021) | CC BY-NC-ND 4.0 | English | Finance | 5000 |
| Fisher (Post et al., 2013) | LDC | English | Conversation | 2000 |
| VoxPopuli (Wang et al., 2021b) | CC BY-NC 4.0 | 23 | Legal | 1800 |
| Googlei18n (Chen et al., 2023a) | Varies | 34 | Variety | 1328 |
| BABEL (IARPA) | IARPA Babel License | 17 | Conversation | 1000 |
| FLEURS (Conneau et al., 2022) | CC BY 4.0 | 102 | News | 1000 |
| KSponSpeech (Bang et al., 2020) | MIT | Korean | Conversation | 970 |
| LibriSpeech (Panayotov et al., 2015) | CC BY 4.0 | English | Audiobook | 960 |
| MagicData (Yang et al., 2022) | CC BY-NC-ND 4.0 | Mandarin | Conversation | 755 |
| TEDLIUM3 (Hernandez et al., 2018) | CC BY-NC-ND 3.0 | English | Talk | 500 |
| Fisher Callhome Spanish (Post et al., 2013) | CC BY-SA 3.0 | 2 | Conversation | 241 |
| VoxForge (VoxForge) | GPL | 8 | Read | 235 |
| AISHELL (Bu et al., 2017) | Apache 2.0 | Mandarin | Read | 200 |
| AIDATATANG (Beijing DataTang Technology Co.) | CC BY-NC-ND 4.0 | Mandarin | Read | 140 |
| AMI (Carletta, 2007) | CC BY 4.0 | English | Meetings | 100 |
| VCTK (Yamagishi et al., 2019) | CC BY 4.0 | English | Read | 25 |

## A. Dataset

For the base 180K hours experiments, we use the exact same corpora as those in OWSM v3.2 (Tian et al., 2024). We emphasize that all of these corpora are *publicly accessible* (although not necessarily purely *open-source* due to some licensing restrictions). In total, this leads to 25 corpora across 151 languages. Following Tian et al. (2024), the target text data is normalized by restoring punctuation and casing. In total, there are 150K hours of data for ASR and 30K hours of data for ST. Details on the license, languages, domain, and size of each corpora are shown in Table 6. A per-language distribution of the 150K hours of ASR data is shown in the third column of Table 13.

To scale to 360K hours, we collect more data from YODAS (Li et al., 2023), which contains 500K hours of speech. However, since the data is crawled from YouTube, the transcripts are very noisy. We therefore obtained and used a clean 180K hour subset of YODAS from Peng et al. (2025), will be made publicly available in the near future. A breakdown of the amount of additional data per language is available in the last column in Table 13.

## B. Training Details

All models use a total effective batch size of 256 utterances and are trained for 675K steps. We use the Adam optimizer (Kingma & Ba, 2015) with a piecewise scheduler (Peng et al., 2024) that linearly warms up the learning rate from 0 to 5.0e-5 in the first 30K steps, 5.0e-5 to 2.0e-4 in the next 30K steps, and finally exponentially decays for the remaining training steps. For the hybrid CTC/attention (Watanabe et al., 2017b) training, we use a CTC weight of 0.3. We use bfloat 16, Flash Attention 2 (Dao, 2024), and DeepSpeed Zero Stage-2 (Rasley et al., 2020; Rajbhandari et al., 2020) to improve training efficiency.

As mentioned in Section 3.2, all OWLS models follow a Transformer (Vaswani et al., 2017) encoder-decoder architecture trained using a hybrid CTC/attention (Graves et al., 2006; Watanabe et al., 2017b) loss. Both the encoder and decoder use

*Table 7.* **Training details for each model size.**

| Params. | Data Hrs. | GPU Type | GPUs per Node | Nodes | Days Training | GPU Hours |
|---|---|---|---|---|---|---|
| 0.25B | 180K | H100 | 8 | 2 | 3 | 1,164 |
| 0.50B | 180K | H100 | 8 | 2 | 4 | 1,512 |
| 1B | 11K | H100 | 8 | 2 | 6 | 2,232 |
| 1B | 22K | H100 | 8 | 3 | 5 | 2,790 |
| 1B | 45K | H100 | 8 | 3 | 5 | 2,790 |
| 1B | 90K | H100 | 8 | 3 | 5 | 2,790 |
| 1B | 180K | H100 | 8 | 3 | 5 | 2,790 |
| 1B | 360K | H200 | 4 | 8 | 7 | 5,120 |
| 2B | 180K | H100 | 8 | 2 | 7 | 2,520 |
| 4B | 180K | H100 | 8 | 3 | 9 | 5,148 |
| 9B | 180K | H100 | 8 | 3 | 15 | 8,472 |
| 18B | 180K | H100 | 8 | 6 | 17 | 19,440 |

sinusoidal absolute positional embeddings (Vaswani et al., 2017). The inputs to the Transformer encoder are 80-dimension log-Mel filterbanks extracted with a frame shift of 10ms, which are then down-sampled 4 times by a stack of convolution layers. The Transformer decoder auto-regressively predicts text tokens, which are pre-segmented with a unigram language model (Kudo, 2018) into a 50K subword vocabulary. We also use Whisper-style training (Radford et al., 2023): all utterances are padded to 30 seconds, and the model is jointly trained to perform language identification, ASR, ST, and timestamp prediction. The exact configurations for each model size are shown in Table 8. We use a mix of A100, H100, and GH200 GPUs for supervised training (Table 7).

*Table 8.* **Architecture hyper-parameter details for each model size.**

| Params. | Enc./Dec. Layers | Hidden Size | FFN Size | Attn. Heads |
|---|---|---|---|---|
| 0.25B | 8 | 768 | 3072 | 16 |
| 0.50B | 16 | 1024 | 4096 | 16 |
| 1B | 32 | 1024 | 4096 | 16 |
| 2B | 16 | 2048 | 8192 | 64 |
| 4B | 36 | 2048 | 8192 | 64 |
| 9B | 39 | 2816 | 11264 | 64 |
| 18B | 64 | 3072 | 12288 | 64 |

## C. Beam Search

*Table 9.* **WER on FLEURS Spanish and Turkish when using greedy search (left) and balancing test-time compute budget (right).**

| Params. | Beam Size | TFLOPS | Spanish WER | Turkish WER |
|---|---|---|---|---|
| 0.25B | 1 / 10 | 1.3 / 48.7 | 27.5 / 22.9 | 82.0 / 69.5 |
| 2B | 1 / 4 | 11.1 / 36.2 | 10.6 / 9.4 | 42.3 / 34.5 |
| 4B | 1 / 2 | 23.4 / 42.3 | 9.0 / 8.4 | 34.5 / 30.6 |
| 9B | 1 | 47.7 | 9.7 | 29.2 |

Table 9 shows additional inference-time compute scaling experiments on the Spanish and Turkish test sets of FLEURS. We use the same beam sizes as the English experiments in Table 2.

## D. Code-Switching

The code-switching evaluation data is collected from Yan et al. (2025). The authors create synthetic code-switching text by taking sentences from 12 non-English languages in FLEURS (Conneau et al., 2022) and randomly swapping in English

translations via dictionary mapping. The swapping is done at the word-level. Bilingual volunteers are then tasked to read the code-switched speech. All volunteers are native speakers in the non-English language and at least fluent in English. The languages to create the code-switched text are Arabic, Czech, Chinese, German, French, Hindi, Japanese, Korean, Portuguese, Russian, Spanish, and Telugu.

## E. Japanese and Taiwanese Chinese Mandarin ASR

This section expands the orthographic analyses results and compares the performance of OWLS on ReazonSpeech Japanese (Yin et al., 2023) and Common Voice Taiwanese Chinese Mandarin (Ardila et al., 2020) against Whisper Large v3 (Radford et al., 2023). The results are shown in Table 10. All OWLS models beyond 4B parameters outperform Whisper Large v3. OWLS 9B achieves the best performance on Reazonspeech with 7.3 CER, less than half of that of Whisper (15.1 CER). OWLS 18B achieves the best performance on Taiwanese Mandarin with a CER of 18.7, while Whisper has a CER of 26.9.

Table 10. ASR performance against SOTA models on Japanese (Reazonspeech) and Taiwan Chinese Mandarin (Common Voice).

| Model | Japanese (ja-jp) | Taiwanese Chinese Mandarin (zh-tw) |
|---|---|---|
| Whisper Large v3 | 15.1 | 26.9 |
| OWLS 0.25B | 14.7 | 50.2 |
| OWLS 0.5B | 9.3 | 39.7 |
| OWLS 1B | 7.8 | 31.0 |
| OWLS 2B | 7.6 | 29.7 |
| OWLS 4B | 7.7 | 24.6 |
| OWLS 9B | **7.3** | 19.1 |
| OWLS 18B | 7.5 | **18.7** |

## F. Mondegreens

As discussed in Section 5.2, mondegreens are cases where a human mishears a phrase in a somewhat semantically coherent manner. For Chinese and Japanese speakers, these are known as 空耳 (kōng'ěr / soramimi). These can either occur within a language ("José, can you see" vs "O say can you see") or across languages ("Bon Appétit" vs "Bone Apple Tea"). We focus on the cross-lingual mondegreen setting, since generating monolingual mondegreens are challenging due to the strength of modern ASR systems. To do this, we first randomly select three low-resource languages (Thai, Afrikaans, and Vietnamese) from FLEURS. We have each model perform ASR inference on these languages, but purposefully input an incorrect English language task tag[4].

For the human evaluation, we have 13 volunteers rate the semantic coherence of the text corresponding to each utterance on a scale from 1 to 5. Scores of 1 indicate completely non-English text or random strings, while scores of 5 correspond to coherent and realistic English words. We filter out all utterances with an average score across all models below 3.0, removing all utterances that are naturally unsuited for creating English mondegreens. Finally, we obtain the average human score for each individual model output, and report the score averaged across all utterances for each model in Table 4. Sample outputs are shown in Table 11.

## G. Contextual Biasing

Previous studies (Peng et al., 2024) have shown that zero-shot contextual biasing is an ability emergent in larger (1B+) ASR models. In this section, we scale the evaluation to the 18B setting. We use the same Librispeech contextual biasing data as Peng et al. (2024); Le et al. (2021), where the model is prompted with a list of true target rare words and distracters. The goal of this task is to reduce the biased WER (B-WER) without degrading the unbiased WER (U-WER). Similar to the results in ST (Section 4.1), we find that small models may encounter catastrophic failures in contextual biasing: the 0.25B model yields a WER of near 97% by frequently outputting blank predictions (Table 12). The 0.5B model also encounters performance degradations upon using contextual biasing prompts, albeit at a less severe magnitude. 1B+ parameter models

---

[4]We initially attempted this evaluation with high-resource non-English languages, but found that models would ignore the incorrect task tag and always transcribe in the original language. We leave further studies of this phenomena to future work.

*Table 11.* **Example mondegreen generations and their corresponding original text.**

| Source | Text |
|---|---|
| Original | Vir daardie rede, als wat jy op die TV sien, het die kante gesny, bo, onder en kante. |
| 0.25B | Dore the rear of the ozvatioctiya fissic. |
| 0.5B | For Dore the Rieda also got the optic fissure. |
| 1B | The order did read as Vatican's affiliate for the first time. |
| 2B | The Daily Director also wrote the optics for his work. |
| 4B | For the order read, also what the optieth is. |
| 9B | The door of the red house was fatty, and the squad was very tired. |
| 18B | For the ordinary, the oasis varies between the oasis and the oasis. |
| Original | Alle burgers van die Vatikaan Stad is Rooms Katoliek. |
| 0.25B | Alabarkers fan diva |
| 0.5B | Alabama cares for the development of the reservation. |
| 1B | allebergers van the valley |
| 2B | Alabama kerrs fan the game. |
| 4B | Alabama, Cars, Fan, Diva. |
| 9B | All the birds catch the worm. |
| 18B | All the workers found the vat. |

are able to better take advantage of the context words, while maintaining U-WER. In fact, only the 9B model is capable of sufficiently maintaining the U-WER while sufficiently lowering the B-WER to get an overall lower WER.

# H. In-Context Learning

Text-based LLMs are capable of few-shot task performance via in-context learning (ICL) from text prompts at inference time. This is generally done by concatenating consecutive examples together, where each example is an input and expected output pair, and feeding the concatenated text as input into a decoder-only causal language model.

We perform ICL for encoder-decoder ASR models in a similar manner, using the same popular formulation introduced by Wang et al. (2024a;b). The encoder is first used to extract embeddings of each speech example in the ICL *training* set, which are averaged across the sequence dimension and cached. This process is summarized in Figure 13. During inference time, we also extract a time-averaged embedding for the input speech and retrieve the $k$ training samples from the cache with the smallest Euclidean distance from the embedding of the test sample. The audio of the retrieved training samples are then concatenated together, with a half-second pause inserted between each sample. Finally, the speech of the input test utterance is appended at the end. This will be used as the encoder input. The decoder input is therefore the concatenation of the retrieved training examples, with a comma inserted between each sample.

## H.1. Quechua Evaluation

Quechua is a low-resource language indigenous to Peru and does not appear in any of the training data that we use. To perform the Quechua ICL evaluation, we use the IWSLT 2024 (Ahmad et al., 2024) version of the Siminchik corpus (Cardenas et al., 2018). We filter out all utterances longer than 7 seconds and split the corpus such that a speaker can only appear in the training or test set. We then further subsample the training set to 150 utterances to reduce compute costs.

*Table 12.* **WER on zero-shot contextual biasing.**

| Params. | test-clean | | | test-other | | |
|---|---|---|---|---|---|---|
| | WER | U-WER | B-WER | WER | U-WER | B-WER |
| 0.25B | 4.05 | 2.76 | 15.08 | 9.62 | 7.33 | 30.82 |
| + biasing | 97.73 | 98.41 | 91.84 | 98.88 | 99.47 | 93.49 |
| 0.50B | 2.65 | 1.77 | 10.22 | 6.61 | 4.85 | 22.97 |
| + biasing | 2.40 | 1.83 | 7.24 | 6.09 | 4.94 | 16.83 |
| 1B | 2.30 | 1.50 | 9.14 | 5.59 | 4.02 | 20.25 |
| + biasing | 2.04 | 1.54 | 6.31 | 5.19 | 4.19 | 14.50 |
| 2B | 2.18 | 1.44 | 8.44 | 5.18 | 3.73 | 18.7 |
| + biasing | 1.98 | 1.50 | 5.98 | 4.63 | 3.70 | 13.23 |
| 4B | 2.03 | 1.37 | 7.60 | 4.65 | 3.33 | 16.90 |
| + biasing | 2.02 | 1.68 | **4.89** | 5.13 | 4.47 | **11.32** |
| 9B | 1.89 | **1.25** | 7.39 | 4.52 | **2.97** | 18.93 |
| + biasing | **1.72** | 1.29 | 5.32 | **4.47** | 3.67 | 11.93 |

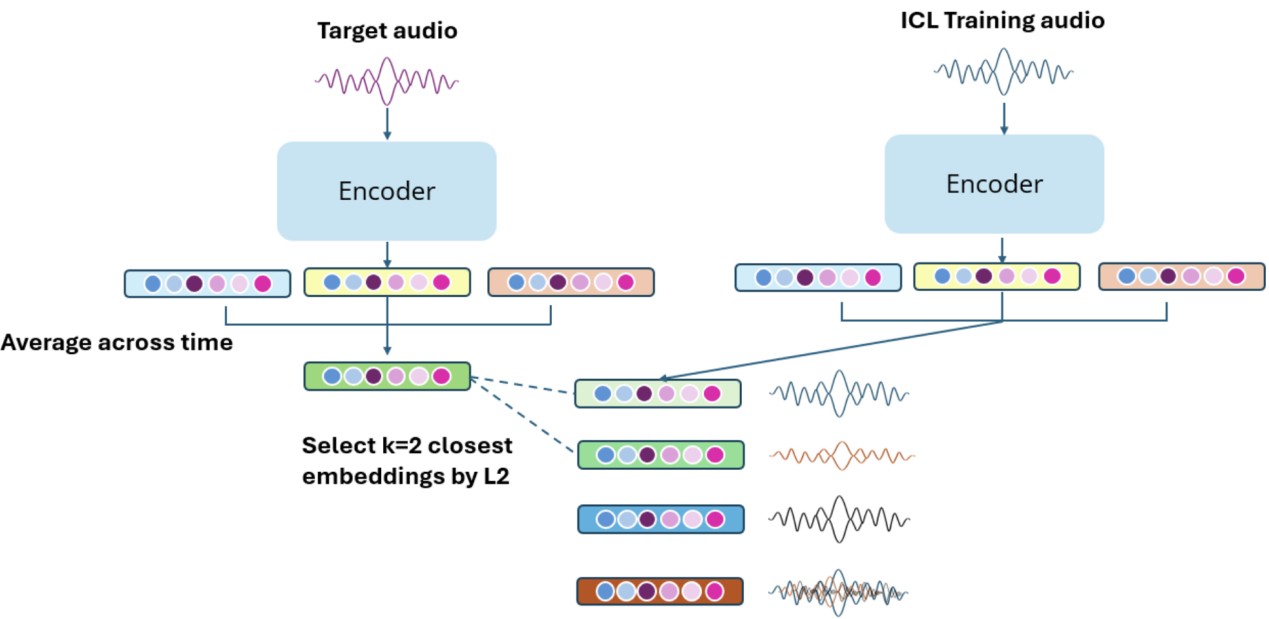

*Figure 13.* The ICL embedding mining process for selecting k in-context examples. For a single test audio input, the encoder extracts an audio embedding of the test audio and all audio samples in the ICL training set. The embeddings are averaged across the sequence dimension. The in-context examples are selected by choosing the top k embeddings with the smallest L2 distance from the test audio embedding.

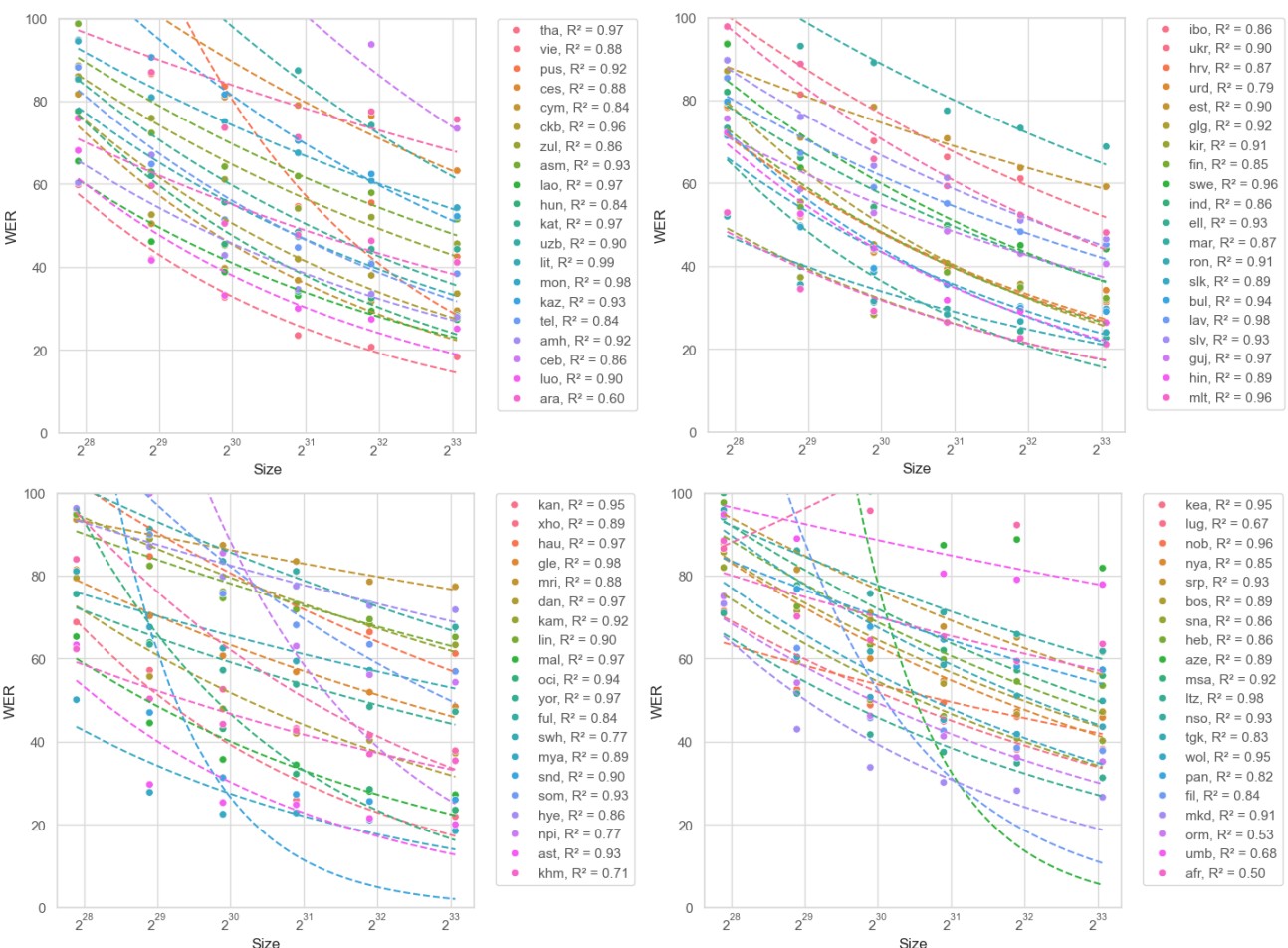

*Figure 14.* **Model scaling laws for all languages in FLEURS.** For almost all languages, WER/CER strongly correlated with the power law w.r.t. model parameter size.

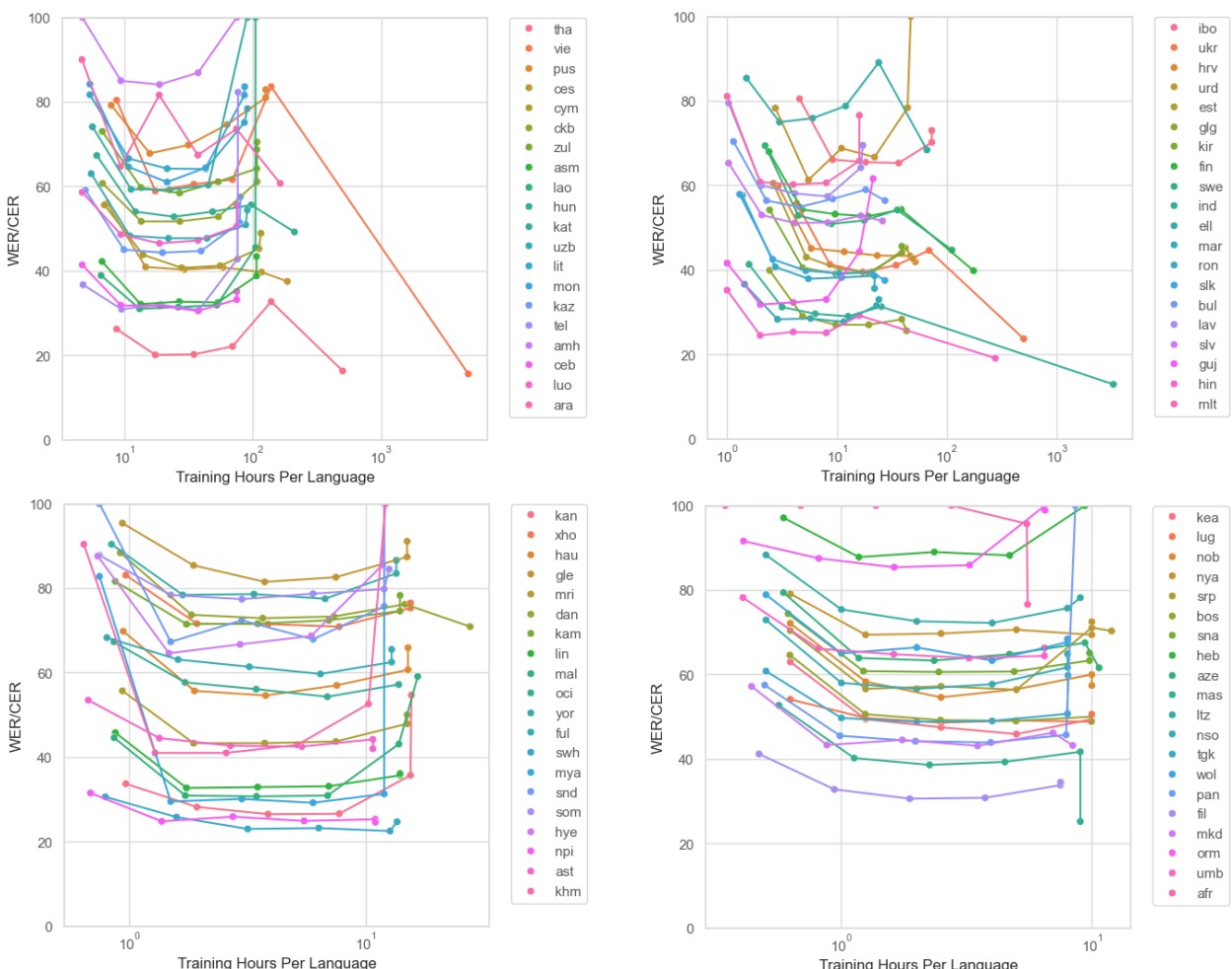

*Figure 15.* **Change in WER/CER when adding more data per language.** For most languages, we observe the same trend as Figure 5: more data with no increase in diversity does not lead to meaningful changes in WER/CER.

*Table 13.* **Amount of ASR training data per language in the OWSM v3.2 180K and YODAS 180K corpora for the top 50 languages in FLEURS.**

| Language | ISO3 Code | OWSM v3.2 Hours | YODAS Hours |
|---|---|---|---|
| English | eng | 73000 | 75000 |
| Russian | rus | 20183 | 15692 |
| Japanese | jpn | 18900 | 1934 |
| Chinese | cmn | 16000 | 176 |
| German | deu | 3700 | 7129 |
| French | fra | 2500 | 8560 |
| Spanish | spa | 2000 | 17344 |
| Catalan | cat | 1996.7 | 37 |
| Dutch | nld | 1700 | 1193 |
| Belarussian | bel | 1319.31 | 0.15 |
| Korean | kor | 1000 | 10890 |
| Italian | ita | 700 | 5553 |
| Bengali | ben | 373.6 | 9.2 |
| Javanese | jav | 372 | 0 |
| Persian | fas | 309 | 30 |
| Polish | pol | 300 | 465 |
| Portuguesse | por | 300 | 10815 |
| Tamil | tam | 296 | 10 |
| Cantonese | yue | 235 | 0 |
| Turkish | tur | 199 | 2322 |
| Thai | tha | 139 | 363 |
| Vietnamese | vie | 139 | 4644 |
| Pastho | pus | 126 | 0.141 |
| Czech | ces | 117 | 69 |
| Welsh | cym | 111.6 | 4.2 |
| Kurdish | ckb | 108 | 0 |
| Zulu | zul | 107.5 | 0 |
| Assamese | asm | 106.878 | 0.06 |
| Lao | lao | 105 | 0.007 |
| Hungarian | hun | 97.3 | 114 |
| Georgian | kat | 90.05 | 1 |
| Uzbek | uzb | 88 | 2.7 |
| Lithuanian | lit | 86 | 5 |
| Mongolian | mon | 86 | 0.42 |
| Kazakh | kaz | 79.18 | 0.9 |
| Telugu | tel | 76 | 0.7 |
| Amharic | amh | 75 | 0.33 |
| Cebuano | ceb | 75 | 0 |
| Luo | luo | 75 | 0 |
| Arabic | ara | 74.6 | 89 |
| Igbo | ibo | 73.017 | 0 |
| Ukranian | ukr | 69 | 433 |
| Croatian | hrv | 46.6 | 5.1 |
| Urdu | urd | 44 | 3 |
| Estonian | est | 42.1 | 7 |
| Galician | glg | 39 | 4 |
| Kyrgyz | kir | 39 | 0 |
| Finnish | fin | 38.5 | 137 |
| Swedish | swe | 35.5 | 76 |
| Indonesian | ind | 25.3 | 3270 |

*Table 14.* **Amount of ASR training data per language in the OWSM v3.2 180K and YODAS 180K corpora for the bottom 50 languages in FLEURS.**

| Language | ISO3 Code | OWSM v3.2 Hours | YODAS Hours |
|---|---|---|---|
| Greek | ell | 24 | 42 |
| Marathi | mar | 23 | 1.1 |
| Romanian | ron | 22 | 0 |
| Slovakian | slk | 20.8 | 6.5 |
| Bulgarian | bul | 18.27 | 9.1 |
| Latvian | lav | 16.5 | 0.704 |
| Slovenian | slv | 16.5 | 9.5 |
| Gujarati | guj | 16 | 5.32 |
| Hindi | hin | 16 | 261 |
| Maltese | mlt | 16 | 0 |
| Kannada | kan | 15.5 | 0.14 |
| Xhosa | xho | 15.5 | 0 |
| Hausa | hau | 15.13 | 0 |
| Irish | gle | 15 | 0 |
| Maori | mri | 15 | 0 |
| Danish | dan | 14.7 | 13 |
| Kamba | kam | 14 | 0 |
| Lingala | lin | 14 | 0 |
| Malayalam | mal | 13.84 | 2.8 |
| Occitan | oci | 13.8 | 0.017 |
| Yoruba | yor | 13.5 | 0 |
| Fulah | ful | 12.9 | 0 |
| Swahili | swh | 12.7 | 0.9 |
| Malaysian | mya | 12 | 0 |
| Sindhi | snd | 12 | 0.009 |
| Somali | som | 12 | 0.6 |
| Armenian | hye | 11.8 | 0.24 |
| Nepali | npi | 11 | 0 |
| Asturian | ast | 10.75 | 0 |
| Cambodian | khm | 10.3 | 1.86 |
| Kabuverdianu | kea | 10 | 0 |
| Luganda | lug | 10 | 0 |
| Norwegian | nob | 10 | 0 |
| Nyanja | nya | 10 | 0 |
| Serbian | srp | 10 | 2 |
| Bosnian | bos | 9.96 | 0 |
| Shona | sna | 9.8 | 0 |
| Hebrew | heb | 9.4 | 0 |
| Azerbaijani | aze | 9.39 | 1.3 |
| Masai | mas | 9 | 0 |
| Luxembourgish | ltz | 8 | 1 |
| Northern Sotho | nso | 8 | 0 |
| Tajik | tgk | 8 | 0.032 |
| Wolo | wol | 8 | 0 |
| Panjabi | pan | 7.9 | 0.7 |
| Filipino | fil | 7.5 | 0 |
| Macedonian | mkd | 7 | 1.4 |
| Oromo | orm | 6.5 | 0 |
| Umbundu | umb | 6.47 | 0 |
| Afrikaans | afr | 5.5 | 0.041 |

*Table 15.* **WER/CER for the top 50 languages in FLEURS by OWLS training data.**

| Languages | 0.25B | 0.50B | 1B | 2B | 4B | 9B | 18B |
|---|---|---|---|---|---|---|---|
| English | 16.8 | 11.8 | 9.7 | 9.5 | 8.5 | 8.5 | **7.7** |
| Russian | 36.4 | 23.8 | 17.5 | 18.5 | 14.7 | 14.8 | **14.5** |
| Japanese | 21.2 | 11.4 | 9.2 | 9.3 | 8.1 | 7.7 | **7.3** |
| Chinese | 26.9 | 17.4 | 13.8 | 13.1 | 12.3 | 11.6 | **10.6** |
| German | 27.0 | 16.6 | 12.7 | 11.7 | 10.2 | 10.0 | **9.5** |
| French | 36.4 | 23.1 | 17.5 | 16.6 | 14.4 | 13.7 | **13.2** |
| Spanish | 27.5 | 15.8 | 11.1 | 10.6 | 9.0 | 9.7 | **9.0** |
| Catalan | 30.5 | 15.2 | 11.4 | 11.1 | 8.8 | **8.6** | 8.8 |
| Dutch | 47.7 | 33.2 | 25.8 | 21.9 | 19.3 | 17.9 | **17.3** |
| Belarussian | 46.0 | 24.2 | 19.1 | 18.5 | **15.8** | 16.2 | 18.2 |
| Korean | 259.2 | 58.8 | 49.4 | 47.7 | 38.6 | 38.5 | **34.2** |
| Italian | 37.9 | 19.5 | 13.9 | 12.2 | **9.8** | 10.0 | **9.8** |
| Bengali | 45.5 | 23.1 | 17.8 | 17.2 | 15 | 13.8 | **13.0** |
| Javanese | 82.2 | 39.2 | 30.6 | 28.8 | 25.7 | 23.6 | **22.7** |
| Persian | 73.0 | 39.0 | 34.0 | 31.1 | 29.3 | 28.9 | **25.6** |
| Polish | 78.2 | 49.0 | 39.1 | 34.7 | 30.0 | 27.7 | **26.5** |
| Portuguesse | 50.1 | 29.6 | 20.5 | 20.5 | 15.7 | **14.1** | 21.1 |
| Tamil | 55.9 | 27.5 | 25.1 | 22.0 | 19.0 | 17.7 | **16.4** |
| Cantonese | 92.5 | 55.1 | 43.9 | 38.0 | 32.7 | 30.0 | **28.4** |
| Turkish | 82.0 | 51.9 | 44.1 | 42.3 | 34.5 | 29.2 | **26.3** |
| Thai | 59.8 | 41.9 | 32.7 | 23.5 | 20.7 | 18.3 | **17.6** |
| Vietnamese | 181.2 | 86.6 | 83.6 | 54.6 | 55.5 | **42.5** | 47.9 |
| Pastho | 114.2 | 108.3 | 81.0 | 79.0 | 76.5 | **63.2** | 64.2 |
| Czech | 81.7 | 50.4 | 39.7 | 36.8 | 31.8 | **29.5** | 36.1 |
| Welsh | 86.0 | 52.6 | 45.2 | 41.9 | 38.0 | **33.6** | 38.9 |
| Kurdish | 88.6 | 75.9 | 61.1 | 54.1 | 52.0 | **45.6** | 49.1 |
| Zulu | 98.7 | 72.3 | 64.2 | 61.9 | 57.9 | **51.5** | 51.9 |
| Assamese | 65.5 | 46.1 | 38.8 | 33.1 | 29.4 | **27.3** | 29.4 |
| Lao | 77.6 | 62.9 | 45.5 | 34.1 | 32.5 | **28.3** | 32.5 |
| Hungarian | 94.8 | 62.0 | 55.6 | 48.4 | **44.3** | **44.3** | 44.7 |
| Georgian | 130.9 | 117.1 | 106.3 | 87.4 | 74.2 | 54.2 | **45.1** |
| Uzbek | 85.3 | 59.6 | 50.9 | 47.3 | 41.0 | 38.4 | **35.2** |
| Lithuanian | 94.5 | 80.9 | 75.1 | 67.5 | 60.8 | 54.3 | **52.7** |
| Mongolian | 117.0 | 90.6 | 81.6 | 70.5 | 62.4 | 52.2 | **48.0** |
| Kazakh | 88.2 | 64.8 | 51.4 | 44.7 | 40.7 | 38.4 | **34.4** |
| Telugu | 60.3 | 67.0 | 42.8 | 34.6 | 33.5 | 27.9 | **26.1** |
| Amharic | 174.9 | 122.4 | 117.1 | 104.9 | 93.7 | 73.4 | **70.9** |
| Cebuano | 68.1 | 41.6 | 33.2 | 30.0 | 27.4 | 25.1 | **24.5** |
| Luo | 75.9 | 59.6 | 50.5 | 47.7 | 46.3 | 41.1 | **39.3** |
| Arabic | 107.5 | 87.0 | 73.6 | 71.3 | 77.5 | 75.6 | **72.1** |
| Igbo | 109.4 | 81.4 | 70.2 | 66.3 | 61.1 | 59.1 | **58.8** |
| Ukranian | 78.3 | 51.9 | 44.6 | 40.5 | 34.6 | 31.3 | **30.2** |
| Croatian | 78.7 | 52.3 | 43.3 | 39.3 | 35.1 | 34.2 | **32.2** |
| Urdu | 93.8 | 71.0 | 78.4 | 70.8 | 63.7 | 59.2 | **58.1** |
| Estonian | 87.1 | 55.5 | 45.2 | 40.7 | 35.7 | 31.8 | **30.5** |
| Galician | 52.7 | 37.3 | 28.3 | 26.4 | 22.8 | 21.0 | **20.1** |
| Kyrgyz | 79.4 | 54.2 | 44.0 | 38.5 | 34.9 | 32.3 | **31.1** |
| Finnish | 93.6 | 63.7 | 54.3 | 49.8 | 45.0 | 44.1 | **42.1** |
| Swedish | 82.0 | 66.1 | 54.2 | 48.9 | 43.1 | 40.7 | **39.1** |
| Indonesian | 73.3 | 39.9 | 31.3 | 29.7 | 24.3 | 22.7 | **22.0** |

*Table 16.* **WER/CER for the bottom 52 languages in FLEURS by OWLS training data.**

| Languages | 0.25B | 0.50B | 1B | 2B | 4B | 9B | 18B |
|---|---|---|---|---|---|---|---|
| Greek | 116.6 | 93.1 | 89.1 | 77.5 | 73.3 | 68.8 | **64.9** |
| Marathi | 52.0 | 35.6 | 31.6 | 28.4 | 26.7 | 24.0 | **21.9** |
| Romanian | 71.5 | 49.7 | 38.6 | 35.5 | 30.3 | 29.7 | **28.7** |
| Slovakian | 79.7 | 49.4 | 39.5 | 35.6 | 29.9 | **29.1** | 29.4 |
| Bulgarian | 85.4 | 67.3 | 59.0 | 55.1 | 48.3 | 45.2 | **43.4** |
| Latvian | 89.7 | 76.0 | 64.2 | 61.3 | 51.0 | 46.5 | **46.0** |
| Slovenian | 75.6 | 58.6 | 52.8 | 48.4 | 43.0 | **40.5** | 42.8 |
| Gujarati | 72.2 | 52.6 | 44.3 | 31.8 | 29.2 | **26.4** | 33.1 |
| Hindi | 52.9 | 34.5 | 29.2 | 26.5 | 22.6 | **21.2** | 22.7 |
| Maltese | 97.8 | 88.8 | 65.8 | 59.3 | 52.3 | **48.1** | 51.7 |
| Kannada | 68.8 | 57.2 | 35.7 | 25.9 | 25.5 | **21.9** | 23.3 |
| Xhosa | 110.5 | 84.7 | 75.3 | 72.5 | 66.4 | **61.2** | 61.2 |
| Hausa | 81.5 | 70.3 | 60.7 | 56.8 | 51.9 | 48.4 | **33.0** |
| Irish | 93.6 | 89.6 | 87.4 | 83.5 | 78.6 | 77.4 | **33.3** |
| Maori | 79.5 | 55.7 | 47.9 | 42.0 | 40.3 | 37.2 | **35.1** |
| Danish | 95.9 | 88.9 | 76.2 | 73.2 | 68.3 | **63.3** | 69.8 |
| Kamba | 94.8 | 82.4 | 74.6 | 71.9 | 69.5 | 65.2 | **64.9** |
| Lingala | 65.3 | 44.5 | 35.7 | 34.4 | 28.0 | **27.2** | 28.8 |
| Malayalam | 100.6 | 63.9 | 43.1 | 32.2 | 28.5 | **23.5** | 26.6 |
| Occitan | 75.6 | 63.4 | 57.2 | 53.8 | 48.4 | **47.2** | 47.8 |
| Yoruba | 104.8 | 91.3 | 83.5 | 81.1 | 73.2 | 67.6 | **52.1** |
| Fulah | 81.1 | 67.6 | 62.5 | 59.4 | 57.2 | 56.9 | **56.0** |
| Swahili | 50.1 | 27.8 | 22.5 | 22.8 | 21.1 | 18.5 | **17.6** |
| Malaysian | 162.8 | 47.0 | 31.3 | 27.3 | 25.6 | 26.0 | **25.6** |
| Sindhi | 126.5 | 90.0 | 75.7 | 68.1 | 63.4 | 56.9 | **54.4** |
| Somali | 96.3 | 87.1 | 79.8 | 77.5 | 72.8 | 71.8 | **70.9** |
| Armenian | 231.7 | 99.9 | 85.5 | 63.0 | 56.1 | 54.3 | **52.1** |
| Nepali | 63.3 | 29.7 | 25.3 | 24.8 | 21.5 | 20.0 | **19.8** |
| Asturian | 62.3 | 50.3 | 44.2 | 43.2 | 37.0 | **35.4** | 37.0 |
| Cambodian | 84.0 | 105.0 | 52.6 | 42.7 | 41.5 | 37.8 | **35.2** |
| Kabuverdianu | 73.1 | 59.3 | 49.4 | 44.9 | **38.1** | 38.2 | 40.6 |
| Luganda | 71.5 | 52.6 | 48.8 | 49.4 | 46.0 | 46.7 | **44.1** |
| Norwegian | 87.8 | 71.5 | 60.0 | 54.7 | 47.2 | 45.8 | **45.3** |
| Nyanja | 102.5 | 81.5 | 69.4 | 67.7 | 65.1 | 61.6 | **60.0** |
| Serbian | 85.7 | 71.5 | 71.1 | 54.0 | **46.5** | 47.2 | 47.1 |
| Bosnian | 82.0 | 59.3 | 50.0 | 46.1 | 40.8 | **40.2** | 40.7 |
| Shona | 97.7 | 72.6 | 63.3 | 59.8 | 54.5 | 53.5 | **51.7** |
| Hebrew | 508.6 | 132.5 | 100.4 | 87.4 | 88.8 | 81.9 | **76.9** |
| Azerbaijani | 100.0 | 77.7 | 67.5 | 62.0 | 57.2 | 55.9 | **53.1** |
| Masai | 71.0 | 51.6 | 41.7 | 37.5 | 34.8 | 31.3 | **30.3** |
| Luxembourgish | 94.2 | 86.1 | 75.7 | 71.2 | 65.9 | 61.8 | **60.2** |
| Northern Sotho | 95.9 | 77.0 | 61.7 | 58.5 | 51.0 | 49.8 | **49.4** |
| Tajik | 86.6 | 60.4 | 50.7 | 45.3 | **41.8** | 43.6 | **41.8** |
| Wolo | 86.7 | 76.8 | 67.7 | 64.6 | 58.1 | 57.3 | **56.7** |
| Panjabi | 172.0 | 62.5 | 45.7 | 42.6 | 38.5 | 37.8 | **35.1** |
| Filipino | 73.3 | 43.0 | 33.8 | 30.2 | 28.2 | 26.6 | **25.5** |
| Macedonian | 75.1 | 54.2 | 46.2 | 41.3 | 36.2 | 35.2 | **32.1** |
| Oromo | 94.8 | 89.0 | 102.4 | 80.5 | 79.1 | 77.9 | **71.1** |
| Umbundu | 88.5 | 70.2 | 64.4 | 65.3 | 59.4 | 63.5 | **59.1** |
| Afrikaans | 86.6 | 102.3 | 95.7 | 133.7 | 92.3 | 152.2 | 92.3 |
| Oriya | 240.4 | 167.8 | 71.4 | 63.0 | 47.9 | 44.8 | **40.1** |
| Icelandic | 108.0 | 102.1 | 91.2 | 85.9 | 77.0 | 74.4 | **72.1** |

