# OpenReview forum: "OWLS: Scaling Laws for Multilingual Speech Recognition and Translation Models"
_ICML.cc/2025/Conference — ICML 2025 poster_

### Official Review · Reviewer_QjdC · 2025-03-13

**Overall Recommendation:** 3

**Summary:**

This paper introduces **OWLS, open-source Whisper-style models for multilingual speech recognition (ASR) and translation tasks**, and releases the trained models. The authors empirically derive scaling laws for multilingual speech processing by training OWLS models at varying scales. Experimental results demonstrate that the large OWLS models follow the scaling laws across different languages and tasks and exhibit the emergent abilities like orthographic understanding or code-switching.

### Update after rebuttal
After going through all the reviews from other reviewers as well as author responses, I'm still concerned with the significance of the paper's findings which are mostly expected. But I like the paper's contribution in scaling ASR models, since there have not been such works that show extensive analyses as this study. Thus, I maintain my score as weak accept.

**Claims And Evidence:**

The paper provides reasonable evidences supporting its claim regarding the difficulties in fitting universal scaling laws for language-agnostic data, which is interesting. However, I'm not quite convinced with the claim related to *ICL ability*. Whisper-style models like OWLS are trained primarily for ASR and translation, not instruction-following. These models do not inherently possess instruction-following capabilities; therefore, it is unreasonable to expect ICL behaviors from them. Also, it is pretty obvious to discover the scaling laws with language-specific model size and scaling data with diversity.

**Essential References Not Discussed:**

None

**Experimental Designs Or Analyses:**

The experimental design is well-structured, providing comprehensive analyses. Particularly noteworthy is the thoughtful experimental setup aimed at investigating emergent abilities.

**Methods And Evaluation Criteria:**

The OWLS training corpus is a simple combination of two existing datasets (OWSM and YODAS), which is not special. However, dividing the scaling analysis clearly into model size, data size, and compute budget is an effective approach.

One limitation is that the method lacks detailed justification for scaling the model specifically up to 18B parameters. While experimenting with large-scale models is valuable, the reasoning behind architectural choices, training configurations, or design principles at this scale is insufficiently elaborated. It remains unclear if the chosen design is optimal in scaling.

**Other Comments Or Suggestions:**

Every figures contain excessively small font sizes, making them difficult to read. Additionally, distinguishing elements solely by color is difficult; varying shapes or increasing font sizes is strongly recommended.

**Other Strengths And Weaknesses:**

A notable strength is the comprehensive scaling-law analysis, particularly as this appears to be among the first efforts to investigate large-scale models (up to 18B) specifically for ASR and AST tasks. The experimental analyses—especially the emergent abilities—are well set-up and insightful.

However, a clear weakness is the minimal methodological novelty, given the dataset merely combines existing corpora and there is not enough grounding on the scalable model design. Moreover, most findings (i.e., better performance as the model, data, or compute scales) are obvious and not significantly groundbreaking.

**Questions For Authors:**

1. The authors noted training using timestamp predictions—are these timestamps available for all datasets? If not, how was missing timestamp information handled?

2. Are there any results of training the encoder other than Whisper-style encoders?

3. In Sec. 5.1, the authors investigated scaling with a fixed computational budget. I'm curious if the authors have also explored model performance and scaling trends when varying beam sizes on large models without budget constraints?

**Relation To Broader Scientific Literature:**

The paper relates to existing literature on ASR and AST models, and more broadly, to LLM scalability. To my knowledge, there hasn't been previous works on comprehensive scaling-law analyses at this large model scale, which is a meaningful contribution to this literature.

**Theoretical Claims:**

Not provided.

---

> ### Author Rebuttal · Authors · 2025-03-31
>
> Thank your for your insights and comments.
>
> >  I'm not quite convinced with the claim related to ICL ability. Whisper-style models like OWLS are trained primarily for ASR and translation, not instruction-following...
>
> While Whisper-style models are not trained for instruction-following, instruction-tuning is not a requirement for models to have the capability to perform ICL, although it will certainly improve their ICL capabilities by minimizing the distribution shift between prompts and pre-training data [1]. Early LLMs, such as GPT2 and GPT3 [2,3], were not instruction-tuned and were only trained on pure language modeling. ICL can be described as a byproduct of auto-regressive modeling, which is how ASR/ST are formulated in auto-regressive encoder-decoder frameworks such as Whisper. ICL is thus a reasonable expectation for the OWLS models because they may be able to use in-context examples (speech-text pairs) as a prior for mapping sounds to words. Examples we provide in our "Strengths And Weaknesses" response to Reviewer ogkW illustrate this capability. [4] also shows preliminary results for this, although we believe that their weaker results are due to the much smaller model size of the tested models.
>
> > One limitation is that the method lacks detailed justification for scaling the model specifically up to 18B parameters...
>
> We chose the upper of 18B primarily due to budget constraints. At 18B, we are still able to perform training without sharding the model parameters across GPUs, which is why we are able to maintain such high training efficiency (training time of 17 days). If we performed model sharding, the expected training time would have been ~2 months, which was unfeasible. Additionally, we were still able to maintain the same mini batch size at 18B without performing gradient accumulation at our compute budget, as gradient accumulation would have also slowed down training significantly. We want to maintain identical batch sizes to minimize experimental variance from the hyper-parameters.
>
> In terms of the model architecture, we equally allocate parameters between the encoder and decoder for two main reasons:
>
> - It is the approach adopted by current SOTA models such as Whisper
> - It was found to be the optimal setting in encoder-decoder scaling for NLP [5,6]
>
> Finally, we chose the Transformer architecture due to its ease of scaling. We experimented with other architectures, such as Conformer, but found it was extremely difficult to tune the hyper-parameters and get the model to converge. This is a similar finding to that of Zhang et al., when they trained Google USM [7]. Other works [8] has shown that the advantages of Conformer diminish as the model size grows. Overall, we believe that our design choices are indeed optimal when considering both effectiveness and efficiency. We will add all of this information in the final draft.
>
> > it is pretty obvious to discover the scaling laws with language-specific model size and scaling data with diversity
>
> > However, a clear weakness is the minimal methodological novelty... Moreover, most findings (i.e., better performance as the model, data, or compute scales) are obvious...
>
> We respectfully argue that many insights of the paper are not predictable. While the notion that a larger model/data/compute leads to better performance is indeed obvious, our results showing that the exact performance improvements w.r.t downstream metrics like WER can be modeled as a scaling law is a novel insight. Many previous scaling law papers model only show scaling laws w.r.t test cross-entropy loss, which is a more predictable but less actionable result for practitioners and researchers. We also want to emphasize that the degree to which performance improves at scale is also non-obvious. For example, En to X ST BLEU increases by a very large factor of 21.8% on average when scaling from 2B (already larger than most SOTA speech models like Whisper Large (1.5B) and Seamless (1.2B)) to 18B parameters, showing that there is much to gain at scale.
>
> > Every figures contain excessively small font sizes, making them difficult to read..
>
> We apologize for the inconvenience. We will make sure to rectify this in future editions.
>
> [1] An Explanation of In-context Learning as Implicit Bayesian Inference (Tie et al., ICLR 2022)
>
> [2] Language Models are Unsupervised Multitask Learners (Radford et al., 2019)
>
> [3] Language Models are Few-Shot Learners (Brown et al., 2020)
>
> [4] Can Whisper perform speech-based in-context learning? (Wang et al., ICASSP 2024)
>
> [5] Scaling Laws for Neural Machine Translation (Ghorbani et al., ICLR 2022)
>
> [6] Exploring the Limits of Transfer Learning with a Unified Text-to-Text Transformer (Raffel et al., 2019)
>
> [7] Hearing the AGI: from GMM-HMM to GPT-4o. (Yu Zhang, 2024). https://youtu.be/pRUrO0x637A.
>
> [8] Revisiting Convolution-free Transformer for Speech Recognition (Hou et al., Interspeech 2024)

---

> > ### Comment · Reviewer_QjdC · 2025-04-04
> >
> > Thanks for the clarification regarding my concerns. The paper exhibits various ASR & AST analyses in terms of scaling laws, which is good. However, my concern still lies at the significance of the paper's findings.
> >
> > > Many previous scaling law papers model only show scaling laws w.r.t test cross-entropy loss, which is a more predictable but less actionable result for practitioners and researchers.
> >
> > I don't see much difference btw cross-entropy loss and WER, which I assume are pretty much correlated. Larger model/data/compute naturally leads to lower cross-entropy loss and lower WER, simultaneously, by a scaling law. I cannot find very surprising results drawn from this. I believe other reviewers share the same concern with mine.
> >
> > Still, I like the paper since it's flawless, sound, and giving insights in scaling ASR models. I suggest the authors put more efforts on experimental details and emphasizing the emergent abilities of OWLS. Also, please add the justifications of the model design choices in the future revision.

---

### Official Review · Reviewer_ogkW · 2025-03-13

**Overall Recommendation:** 3

**Summary:**

This paper investigates the effect of model size and dataset size on multilingual Automatic Speech Recognition (ASR) and Speech Translation (ST) tasks for 150 languages. The model sizes vary from 0.25B to 18B parameters. The WER vs. size curves are fitted into power law functions and the correlations are reported in terms of R^2 value. In addition, the paper test the trained models on unseen tasks or languages to show the emerging capabilities of these large multilingual models.
Experiments are performed on up to 360k hours of speech across 150 languages. General trends are intuitive in that larger model sizes and larger data sizes lead to lower WER and higher BLEU scores.

## Update after rebuttal
The paper will benefit from some revision to clarify some of the points mentioned in the rebuttal comment. I would like to keep my initial score.

**Claims And Evidence:**

Claims that are somewhat arguable:

1) Claim: "Larger models can mitigate bias and improve the fairness of speech technologies.". Argument: This is a very generic statement that may or may not be true depending on how you define fairness. In this context, the authors might have meant language coverage and fairness due to have reasonably well ASR model for many languages. However, this is a very narrow definition. There could be multiple dimensions in which the fairness of ASR models can be analyzed and only the analyses in this paper are not sufficient to make such a big fairness claim.

2) I could not quite follow how Section 4.3. tries to estimate the final checkpoint WER from the initial checkpoint WER, and also how this is reflected in Fig. 9. Hence, I could not check the validity of the following: "one can reasonably predict the final WER of the model given the WERs of initial checkpoints."

3) Code-switching Fig. 11. looks interesting, it seems that there are two clusters of languages, low CER ones with a clearer pattern and high CER languages with non-uniform patterns over model size. In the text, some of the difference is attributed to the character set of these languages En - PT being Latin-only versus En- ZH using different characters. However, the results seem a little confusing. For example, why En - AR or En - RU does not suffer from a similar problem?

**Essential References Not Discussed:**

NA

**Experimental Designs Or Analyses:**

Experiments look valid.

**Methods And Evaluation Criteria:**

Yes

**Other Comments Or Suggestions:**

1) Fig. 3 and Fig. 5, did you mean the Tamil language when the figure says Tami?

2) What is PN-CER in Fig. 10 caption? Is it N-CER which was mentioned in the text?

**Other Strengths And Weaknesses:**

- Strength: OWLS can be useful for both ASR and ST community with the analysis it provides. Once the code and the setup is available, it might be helpful for researchers, too.

- Weaknesses: Some arguments are over-generalized, but they do not always hold. For example,
- Fig. 4, was there an experimental issue in the AMI (or Tedlium, cannot quite tell from the color) ASR runs?
- The generic fairness claim (discussed above)
- In-context learning is happening to some extent but the given example may not be sufficient to claim this ability.

**Questions For Authors:**

1) Could you please explain why WER/CER is increasing with increasing data after a certain point in Fig. 5?

2) Could you please rephrase the fairness claim in Section 4.1. Multilingual ASR?

3) Fig. 4., could you please explain why the topmost curve is fluctuating? Also in this figure, it is hard to tell if it AMI or Tedlium or Gigaspeech (the legend font size makes it difficult).

4) How does Section 4.3. estimate the final checkpoint WER from the initial checkpoint WER? How this is reflected in Fig. 9?

5) Code-switching Fig. 11. looks interesting, it seems that there are two clusters of languages, low CER ones with a clearer pattern and high CER languages with non-uniform patterns over model size. In the text, some of the difference is attributed to the character set of these languages En - PT being Latin-only versus En- ZH using different characters. However, the results seem a little confusing. For example, why En - AR or En - RU does not suffer from a similar problem?

**Relation To Broader Scientific Literature:**

The paper investigates large and extra large ASR and ST models in the multilingual scenario. Especially, the differences from 1B to 9B or 18B model that we have not empirically seen in previous studies can lead to some additional research questions for serving ASR model to low-resourced languages.

Once the source code and the models are released, this OWLS suite can be used to setup baselines for other studies easily which would be valuable.

**Theoretical Claims:**

NA

---

> ### Author Rebuttal · Authors · 2025-03-31
>
> Thank your for your insights and comments. We organize our response by section:
>
> ## Supplementary Material
>
> We believe that we have all the possible information for the embedding extraction process in the appendix, although we acknowledge that the description may be unclear and hard to follow through text alone. For further clarity, we will add the following figures to the manuscript to better visualize the embedding extraction method: https://anonymous.4open.science/r/owls-samples-13A3/icl_figure.md . The code to perform ICL will also be released for reproducibility.
>
> ## Claims And Evidence:
>
> 1. We understand the reviewer’s concern. While our definition was indeed focusing on the fact that scaling leads to a better model for many languages, we do acknowledge that this definition may be too narrow. We will instead rephrase the fairness claim to “Scaling to larger models can improve ASR performance across the board, in both low and high resource languages, with significant improvements in the former category.”
>
> 2. We perform inference with the model after 15K, 30K, 60K, 120K, 240K, 480K training steps, and use these datapoints to fit a power-law function. We then extrapolate the power law to 675K steps, which is where the final checkpoint is taken from, and compare it to the true WER of the final checkpoint. In Fig 9, each datapoint corresponds to each checkpoint that we evaluated (note that some early checkpoints are not visible since we cut off the chart at 100 WER). The training FLOPS of each checkpoint is then calculated by profiling each model size for a few steps and scaling the result to 15K-675K steps. We will adjust the manuscript to make these details more clear.
>
> 3. We also arrived at the conclusion that the clusters are largely distinguished by character set. However, we believe that linguistic similarity likely also plays a role. Russian, despite using Cyrillic letters, is more linguistically similar to English and the bottom cluster languages (which are all European), than the top cluster languages. We believe there may be some miscommunication for Arabic due to our presentation format. Arabic is the red line in the middle of the top cluster and certainly suffers from higher CERs. We will update the chart with different shapes for each language to make it less confusing.
>
> ## Other Strengths And Weaknesses
>
> > Fig. 4, was there an experimental issue in the AMI (or Tedlium, cannot quite tell from the color) ASR runs?
>
> We addressed below in our response to Question 3.
>
> > In-context learning is happening to some extent but the given example may not be sufficient to claim this ability.
>
> We have included a few samples on example ICL generations https://anonymous.4open.science/r/owls-samples-C53F/README.md , which should better illustrate the improvements in ASR output as the number of in-context examples increase.  The most obvious change is from pure zero-shot (k=0) to on-shot ICL (k=1), where the model changes from outputting a completely wrong language / character set to text that looks like Quechua. With more examples, the models also start to learn more complex phone-to-orthographic mapping ("ai" -> "ay") and white space placement ("ruraikunatukunapa -> ruraikuna tukunapa").
>
> ## Other Comments Or Suggestions
>
> 1. Yes for both. Apologies for the mistake. We will fix this typo.
> 2. Yes, the caption should also read PN-CER. We will fix this typo.
>
> ## Questions For Authors
>
> 1. We believe that the large increases in WER/CER are due to two main factors: 1) domain mismatches between the additional YouTube data and FLEURS evaluation data and 2) interference from similar languages/dialects (ie. Chinese and Cantonese).
>
> 2. Addressed above in "Claims and Evidence" point 1.
>
> 3. The fluctuation is in AMI, where the data is largely spontaneous and conversational, as the audio is sourced from business meetings. Since our training data is largely read speech and multilingual, it can be expected that there will be some small fluctuations on individual test sets despite average performance improving with scale. We will make the figure easier to read by using different shapes for the points of each dataset.
>
> 4. Addressed above in "Claims and Evidence" point 2.
>
> 5. Addressed above in "Claims and Evidence" point 3.

---

### Official Review · Reviewer_MsZt · 2025-03-16

**Overall Recommendation:** 2

**Summary:**

The paper introduces OWLS, a suite of multilingual speech recognition and translation models ranging from 0.25B to 18B parameters. It systematically studies scaling laws for speech tasks, demonstrating how model, data, and compute scaling impact performance. The paper claims that larger models improve low-resource language performance and exhibit emergent abilities, and it presents empirical scaling laws derived from experiments.

## update after rebuttal

The authors’ response has addressed several of my initial concerns; however, it does not substantially change my overall evaluation of the work. The paper primarily focuses on scaling a standard ASR model rather than introducing novel methodological contributions or architectural innovations.

While I acknowledge the significance of the presented experiments to the ASR community, the results neither revealed surprising findings nor provided sufficient new insights. I was particularly hoping for a more in-depth and reflective discussion on how the Audio-LLM paradigm contrasts with traditional ASR approaches—an aspect that remains unaddressed in the current version.

After reviewing comments from other reviewers, I noted that similar concerns were shared. As such, I will maintain my original evaluation.

**Claims And Evidence:**

The claims about scaling benefits (especially for low-resource languages) are well-supported by experimental results.
The study effectively demonstrates the impact of scaling on WER and BLEU scores across different settings.
Weakness:
While the paper suggests that emergent abilities arise with larger models, some results (e.g., code-switching improvements) appear inconsistent across languages. More justification is needed.

**Essential References Not Discussed:**

The paper does not discuss some recent multilingual speech models (e.g., SeamlessM4T and MMS from Meta).

**Experimental Designs Or Analyses:**

The experiments are well-controlled, and the choice of different model sizes is appropriate. The scaling trends in WER and BLEU are insightful, but the role of domain mismatch in dataset expansion (e.g., adding YODAS data) should be further explored. The analysis of emergent abilities is interesting but would be stronger with qualitative examples.

**Methods And Evaluation Criteria:**

The methodology is well-structured, following systematic scaling experiments.
The use of publicly available data enhances reproducibility, but some details about data preprocessing are missing.

**Other Comments Or Suggestions:**

Correct the legend in Figure 1: The blue color in the legend is incorrectly assigned to Whisper, whereas the caption states that it represents OWLS. Ensure that the legend accurately reflects the intended model categorization to avoid confusion.

**Other Strengths And Weaknesses:**

Strengths:

* The open-source approach enhances reproducibility.

* The paper provides a comprehensive scaling analysis for speech models.

* Strong empirical results demonstrate the model’s efficiency.

Weaknesses:

My primary concern is that the performance of this work appears relatively uncompetitive compared to recent advancements in Audio-LLMs, such as Qwen2-Audio. While this paper primarily focuses on Automatic Speech Recognition (ASR) and Speech Translation (ST), Audio-LLMs support a broader range of tasks and offer wider application scenarios.

I acknowledge that a notable strength of this study lies in its efficiency, achieved through hybrid CTC/Attention training. However, it remains uncertain whether this approach represents the most optimal solution by current standards. Additionally, the paper does not provide a direct comparison with other efficiency-focused models, such as SenseVoice, which raises concerns about the completeness of the evaluation.

Therefore, despite the extensive experiments conducted, the paper lacks a compelling demonstration of its results, which limits its overall impact.

**Questions For Authors:**

1. How does OWLS compare to existing large-scale models like SeamlessM4T and SenseVoice in terms of efficiency?

2. Have you tested the generalization of OWLS on out-of-domain data beyond the current benchmarks?

3. How does the training cost of OWLS compare to other state-of-the-art multilingual speech models?

**Relation To Broader Scientific Literature:**

The paper situates itself well within prior scaling laws research in NLP and speech.

**Theoretical Claims:**

The paper effectively extends scaling laws to speech tasks, building on prior work in text and vision. The derivation of scaling laws is reasonable, but it would benefit from a more formal theoretical justification.
Some empirical observations (e.g., emergent abilities) could be further contextualized within existing theories.

---

> ### Author Rebuttal · Authors · 2025-03-31
>
> Thank you for the review and insights.
>
> > some details about data preprocessing are missing
>
> Reviewer uTgG raised similar concerns. Please refer to our response to their question 3.
>
> > code-switching improvements appear inconsistent across languages.
>
> Since we are performing multilingual multi-domain training and evaluation, some per-language fluctuation is expected. However, the average code-switching CER clearly decreases with increases in model size. We will better illustrate this by including a line for the average performance, along with a table with raw scores in the appendix.
>
> > the role of domain mismatch in dataset expansion should be further explored
>
> While we agree that the domain mismatch should be further explored (and plan to do so in future work), we believe that further experiments are outside this paper's scope, which is focused on multilingual neural scaling rather than studying the role of data domains. The domain shift finding is a consequence of doing such large scaling analyses, rather than a hypothesis we explicitly designed an experiment to test.
>
> > The analysis of emergent abilities would be stronger with qualitative examples.
>
> Table 9 includes example mondegreen generations. We have uploaded the accompanying audio for reference in https://anonymous.4open.science/w/owls-samples-88B2/ . We also added sample generations for ICL (Table 5): https://anonymous.4open.science/r/owls-samples-C53F/README.md . We will include these in the final manuscript.
>
> > More details on hyperparameter selection and convergence criteria
>
> We use the same hyper-parameters as OWSM v3.1, which generalized to all model sizes. We did not have explicit convergence criteria, but instead cut off training after 675K steps for fairness and budgeting. All models appear to have converged by this point in both validation and training loss. We will include convergence plots in the final draft.
>
> > does not discuss some recent multilingual speech models  (SeamlessM4T and MMS)
>
> We reference Seamless in 2.3 and discuss how they differ from Whisper-style models. We also reference MMS at a high-level in 2.3, but will update the draft to make more direct comparisons. WER comparisons with Seamless are shown in the able below.
>
> > performance appears relatively uncompetitive compared to Audio-LLMs, such as Qwen2Audio...does not provide a direct comparison with other efficiency-focused models, such as SenseVoice
>
> Below are comparisons with these models for 3 languages using FLEURS and non-FLEURS test sets. Chinese/Japanese use CER, English uses WER. We use the official reported scores when possible, designated by a *, otherwise we run inference ourselves. Since SenseVoice-L is not publicly available, we only include their official results.
>
> |Model| FLEURS zh | FLEURS en | FLEURS ja | AISHELL 1 zh | test-clean en | Reazon ja |
> |-|-|-|-|-|-|-|
> |OWLS 1B|13.8 | 9.7 | 9.2 | 6.2 | 2.3 | 7.8
> |OWLS 9B|11.6 | 8.5 | 7.7 | 4.8 | 1.9 | 7.3
> |OWLS 18B|10.6 | 7.7 | 7.2 | 4.8 | 2.0 | 7.5
> |OWLS 18B v2|10.1 | 6.8 | 6.7 | 4.8 | 2 | 7.2
> |Qwen2Audio|7.5*|9.4| 20.1 | 8.7 | 1.6* | 50.0
> |SenseVoice S|9.6|10.3| 13.1 | 3.0* | 3.2 *| 37.1
> |SenseVoice L||||2.1*|2.6*
> |Seamless Medium|15.7*|8.3*|15.9*|9.6|4.2|34.9
> |Seamless Large| 17.0*|7.3*|17.6*|8.7|3.7|36.6
>
> OWLS models are competitive with Qwen2Audio:  OWLS 18B v2 outperforms QwenAudio2 on 4/6 test sets. We note that we were also unable to reproduce the results in the Qwen2Audio paper, since they did not release their prompts (we obtained a CER of 14.2 on FLEURS zh with Qwen2Audio, while official is 7.5 CER), further emphasizing the importance of the open nature of the OWLS suite. We found that Qwen2Audio was more susceptible to hallucinations compared to the other models, which is in line with previous studies on LLMs for ASR [1].
>
> ## Questions
>
> 1. OWLS is similar to Seamless and SenseVoice-L in terms of algorithmic efficiency, since they're all encoder-decoders.
>
> 2. All benchmarks we use are in the manuscript. Code-switching and Quechua are out-of-domain.
>
> 3. While it is difficult to compare the training cost to all SOTA models due to the proprietary nature of training details, the below table shows a comparison for the models where the information is available:
>
> | Model | Parameters | GPUs | Training Hours |
> |-|-|-|-|
> |Canary|1B| 128xA100 80GB |48
> |OWSM v3 |900M| 64xA100 40GB |240
> |OWSM v3.1|1B| 64xA100 40GB |384
> |OWSM-CTC |1B| 64xA100 40GB|300
> |OWLS |1B|24xH100 80GB |116
> |OWLS |18B| 48xH100 80GB |405
>
> Note that training hours are not fully comparable due to differences in GPU and cluster environment. Canary also uses a pre-trained speech encoder to decrease the training time, which is not considered in the final budget. Nevertheless, we believe the OWLS training is well-optimized, as we had a limited compute budget. All of these optimizations will be open-sourced.
>
> [1] Performance evaluation of SLAM-ASR: The Good, the Bad, the Ugly, and the Way Forward. (Kumar et al., ICASSP 2025 SALMA Workshop).

---

### Official Review · Reviewer_A8Vp · 2025-03-19

**Overall Recommendation:** 4

**Summary:**

This paper investigates scaling laws for multilingual, multi-task speech-to-text models.

To achieve this, the authors introduce OWLS, a collection of ASR/ST models ranging from 0.25B to 18B parameters, with the 18B model being the largest publicly known ASR/ST model to date.

The study examines three dimensions of scaling, model size, data size, and compute budget, to understand how each factor influences performance. Additionally, the paper highlights emergent abilities that only appear when the model reaches a certain scale.

The OWLS project is fully open-sourced, with the training code, logs, and checkpoints set to be released to facilitate reproducibility and further research.

**Claims And Evidence:**

This paper makes several key claims:

1. Scaling Laws: The most significant claim is that the performance of multilingual ASR/ST models follows a (language-specific) scaling law. This is well-supported by Figures 3, 6, and 7, which demonstrate a consistent relationship between model size, data size, and performance metrics.
2. Benefits for Low-Resource Languages: The paper also claims that larger multilingual ASR models improve performance for low-resource languages by reducing WER. This is generally well-supported by the results. But I'm not sure how the paper demonstrates that it "mitigates bias and improves the fairness of speech technologies." Without further analysis, this claim seems overstated.
3. Emergent Abilities: The section on emergent abilities could be clearer, particularly the discussion of mondegreen phenomena.
4. In-Context Learning (ICL): The ICL results appear inconsistent, lacking a clear trend across different model sizes and settings.

**Essential References Not Discussed:**

Not that I am aware of.

**Experimental Designs Or Analyses:**

Yes.

For the Mondegreen section, I am not particularly sure how PPL/MOS relates to ASR performance, or in other words, how they are useful. Does this phenomenon actually help reduce WER/CER, or is it simply an artifact of large-scale training? If not, why is this ability important enough to be discussed in this paper?

**Methods And Evaluation Criteria:**

Yes, WER/CER and BLEU are both standard criteria for measuring ASR and ST performance.

**Other Comments Or Suggestions:**

No.

**Other Strengths And Weaknesses:**

This paper is well-written and easy to read. One minor comment is that the colors used in the figure are very friendly to color-blind people.

**Questions For Authors:**

See above.

**Relation To Broader Scientific Literature:**

This paper shows that multilingual ASR/ST models improve predictably with scale, especially for low-resource languages. It also explores emergent abilities like mondegreens and code-switching, though these findings need stronger proof. Additionally, it tests In-Context Learning (ICL) in ASR, but the results are unclear. Overall, the paper applies scaling laws to speech models and raises interesting questions about model behavior at large scales.

**Theoretical Claims:**

Not applicable, as this is an empirical paper.

---

> ### Author Rebuttal · Authors · 2025-03-31
>
> Thank your for your insights and comments. We organize our response by section:
>
> ## Claims And Evidence:
>
> 2. We understand the reviewer’s concern. While our definition was indeed focusing on the fact that scaling leads to a better model for many languages, we do acknowledge that this definition may be too narrow. We will instead rephrase the fairness claim to “Scaling to larger models can improve ASR performance across the board, in both low and high resource languages, with significant improvements in the former category.”
>
> 3. We have included the accompanying audio for the mondegreen samples in Appendix Table 9 here:  https://anonymous.4open.science/w/owls-samples-88B2/ , which we hope better illustrate the phenomena. If there any particular items that can be further clarified, we will be happy to make those changes as well.
>
> 4. We have included a few samples on example ICL generations https://anonymous.4open.science/r/owls-samples-C53F/README.md , which should better illustrate the improvements in ASR output as the number of in-context examples increase. While there is not a clear monotonic trend, we note that there are 3 clear clusters in ICL CER that are aligned with model size. 0.25B and 0.5B both have ~33.8 CER, 1B to 4B all have ~31.5 CER, and 9B to 18B have ~27.7 CER.
>
> ## Experimental Designs Or Analyses:
>
> > For the Mondegreen section, I am not particularly sure how PPL/MOS relates to ASR performance, or in other words, how they are useful. Does this phenomenon actually help reduce WER/CER, or is it simply an artifact of large-scale training? If not, why is this ability important enough to be discussed in this paper?
>
> PPL/MOS do not have a relationship with ASR performance, since the only goal of these metrics is to measure the English coherence of the mondegreen generation. We note that this phenomenon likely does not have any impact on overall WER/CER, and is probably indeed an artifact of scaling. However, we believe that it is important to discuss and present this evaluation, because it allows researchers to better understand the properties of neural networks that emerge at scale, and how it relates to the human perceptions of spoken language. By including this evaluation, we can show that there is much we do not understand about ASR models and how they scale, opening up potential new areas of research that were previously not possible due to lack of access to such models.

---

### Official Review · Reviewer_uTgG · 2025-03-20

**Overall Recommendation:** 3

**Summary:**

This paper empirically evaluates the scaling law for speech recognition and translation in terms of training data, model size, and compute cost, using a total of 350K hours of multilingual training data.

**Claims And Evidence:**

Yes

**Essential References Not Discussed:**

The reference looks good to me.

**Experimental Designs Or Analyses:**

Yes

**Methods And Evaluation Criteria:**

N/A

**Other Comments Or Suggestions:**

See below

**Other Strengths And Weaknesses:**

Strength
  This paper presents a valuable study, providing a comprehensive analysis of the effects of data scale, model size, and computational cost on speech recognition and translation. The extensive experimental evaluation is highly appreciated.

Weakness
  The paper does not introduce novel ideas or algorithms but instead presents a series of experiments. Moreover, most of the conclusions are predictable, limiting the overall insight gained from the study. Additionally, I have concerns about the analysis, as the transcriptions are sourced from YouTube, and their accuracy remains uncertain, which might affect the conclusion drawn in the paper.

Given the paper’s clear strengths and potential weaknesses, I would rate it as a weak acceptance.

**Questions For Authors:**

1. The conclusions drawn from the study may not be entirely reliable due to the noisy nature of the training data. Scaling trends observed here might not hold when using large-scale, high-quality transcribed audio data. This could also explain the discrepancy in Figure 2, whether the authors inspect the relationship between WER and the amount of training data that does not fully align with expected scaling laws.
2. The experimental results largely align with expectations and do not provide particularly surprising insights. E.g. the statement, “while parameter scaling can significantly improve ST performance, it cannot overcome cases where there is inherently insufficient data to learn the task”, is intuitive and aligns with prior expectations.
3. The paper does not specify the quality of transcriptions used for speech recognition and translation. The authors should describe how the transcriptions were obtained for both Yodas and OWSR datasets, as transcription quality significantly impacts scaling trends.
4. Model Size in Table 1: In Table 1, listing the model sizes of Canary and Whisper would provide better clarity. I assume the authors are using Whisper Large v3—please confirm.
5. The relevance of Table 2 is unclear. Given that WER is generally not highly sensitive to beam size, I would expect minimal variations. If this assumption is incorrect, the authors should provide supporting WER values to demonstrate the impact of beam size.
6. Since this is a multilingual ASR model, observations based on a single language may not generalize across different languages. The authors should clarify whether the observed trends are consistent across multiple languages.

**Relation To Broader Scientific Literature:**

This paper investigates the scaling law for speech, which aligns with the scaling law observed in text data for large language models.

**Theoretical Claims:**

This paper is primarily experiment-driven, this study does not present strong theoretical claims.

---

> ### Author Rebuttal · Authors · 2025-03-31
>
> Thank you for your valuable insights.
>
> > most of the conclusions are predictable, limiting the overall insight gained from the study
>
> We respectfully clarify that many of our insights are not predictable. While the notion that a larger model leads to better performance is indeed obvious, showing that exact performance improvements w.r.t downstream metrics like WER can be modeled as a scaling law is a novel insight. Many previous scaling law papers model only show scaling laws w.r.t test cross-entropy loss, which is a more predictable but less actionable result for researchers.
>
> > concerns about the analysis, as the transcriptions are sourced from YouTube, and their accuracy remains uncertain...  Scaling trends observed here might not hold when using large-scale, high-quality transcribed audio data.
>
> All of our evaluation results are performed on standard benchmarks with high-quality transcriptions, such as FLEURS or CoVoST, commonly used in evaluating many large-scale models such as [1,2]. We only use YouTube transcriptions for helping train two models in this paper (OWSL 1B and 18B 360K), which means scaling curves in figures 1-4,6,7, and 9 are not affected.
>
> ## Questions
>
> 1. Most of our training data is high quality [4], as it comes from common academic benchmarks. We only use noisier transcriptions for OWLS 1B 360K and OWLS 18B 360K. The purpose of Figure 2 is to show scaling patterns across all languages, and that it is unfeasible to fit a language-agnostic universal scaling law. This is a reasonable expectation, since some languages are naturally more difficult to model for ASR than others (i.e Spanish vs Chinese) [5], so they will require more training data. We acknowledge that this point was not clearly made in the paper, and will better word it in the future drafts.
>
> 2. While we believe that the statement is indeed true for traditional ASR/ST models, we respectfully argue that such intuitions may not hold true for large-scale models like OWLS 18B. For example, [6] found that LLMs only require a small amount of leaked bilingual data to learn text translation. It is thus not completely out of the question that a sufficiently large ASR/ST model may be able to overcome traditional data limitations. Due to the potential openness of this question, we believe that this result is valuable evidence and proof to improve the scientific community’s understanding of large-scale models.
>
> 3. A breakdown of all datasets is found in Appendix A/Table 6. The OWSM datasets are high quality [4], as they come from common academic benchmarks, such as LibriSpeech, VoxPopuli, MuST-C, and CoVoST. For YODAS, we obtain a cleaner subset from the dataset authors. They use text language identification on the transcripts (to check that the text/audio language is aligned) and CTC segmentation on the audio with a pre-trained OWSM-CTC [7] model. The worst 10% of the utterances by CTC segmentation score are then discarded. We will make this information more apparent in the main paper body in future drafts.
>
> 4. We apologize for the oversight. Canary is 1B and Whisper is 1.5B. We are using Whisper Large V3, and will make it more clear in future drafts.
>
> 5. We ran each model in Table 2 on LibriSpeech test-other with beam size=1, and compare the results with that of Table 2 below, showing that WER is indeed sensitive to beam size. We will include this analysis in future drafts.
>
> | Model Size | Beam Size | WER |
> |-|-|-|
> | 0.25B | 1 |  9.6
> | 0.25B | 10 |  8.3
> | 2B | 1 |  5.2
> | 2B | 4 |  4.7
> | 4B | 1 |  4.6
> | 4B | 2 |  4.5
> | 9B | 1 |  4.5
> | 9B | 2 |  4.5
>
> 6. We can assert that the observed trends are consistent across multiple languages, as we perform multilingual evaluations in almost every figure/table.The only exception is beam search (Table 2), which we will rectify by including the below results on FLEURS for Spanish and Turkish. Larger models with smaller beam sizes have better WER than smaller models with larger beam sizes. The only exception is 4B and 9B, where 4B already outperforms 9B with beam size=1.
>
> | Model Size | Beam Size | Spanish | Turkish |
> |-|-|-|-|
> | 0.25B | 1 |27.5|82.0
> | 0.25B | 10 |22.9|69.5
> | 2B | 1 | 10.6|42.3
> | 2B | 4 | 9.4|34.5
> | 4B | 1 | 9.0|34.5
> | 4B | 2 | 8.4|30.6
> | 9B | 1 | 9.7|29.2
>
> [1] Robust Speech Recognition via Large-Scale Weak Supervision (Radford et al., ICML 2023).
>
> [2] Gemini: A Family of Highly Capable Multimodal Models (Gemini Team., 2024).
>
> [4] On the effects of heterogeneous data sources on speech-to-text foundation models (Tian et al., Interspeech 2024).
>
> [5] Language Complexity and Speech Recognition Accuracy: Orthographic Complexity Hurts, Phonological Complexity Doesn't. (Taguchi et al., 2024).
>
> [6] Searching for Needles in a Haystack: On the Role of Incidental Bilingualism in PaLM’s Translation Capability (Briakou et al., ACL 2023).
>
> [7] OWSM-CTC: An Open Encoder-Only Speech Foundation Model for Speech Recognition, Translation, and Language Identification (Peng et al., ACL 2024).

---

### Decision · Program_Chairs · 2025-05-01

**Decision:**

Accept (poster)

**Comment:**

This paper introduces OWLS, a suite of multilingual speech recognition and translation models with various sizes (from 250M to 18B parameters). These models are trained using up to 360k hours (OWSM+YODAS) multilingual speech covering 150 languages. The authors systematically investigate the scaling behavior of multilingal speech recogntion and translation with respect to data size, model size and compute budget. In addition, the authors also investigate the emergent capabilities of large models such as orthographic understanding,
code-switching and mondegreens. Overall, the topic under investigation is important and the work is interesting. It is among the first investigations in this line of research at this scale which may have its value to the speech community. The rebuttal clears up most of the concerns and questions raised by the reviewers. There are some lingering concerns though. For example, the paper is basically experiment heavy without introducing any novel ideas or techniques.